# Bidirectional wakes over complex terrain using SCADA data and wake models

Nanako Sasanuma[1], Akihiro Honda[2], Christian Bak[3], Niels Troldborg[3], Mac Gaunaa[3] and Niels Morten Nielsen[3], Teruhisa Shimada[1]

[1]Graduate School of Science and Technology, Hirosaki University, Bunkyo−cho 3, Hirosaki, 036−8561, Japan
[2]Aomori Public University, Department of Management and Economics, Aomori, Aomori, Japan, 030-0196, Japan
[3]DTU Wind and Energy Systems, Technical University of Denmark, Roskilde, 4000, Denmark

*Correspondence to*: Nanako Sasanuma (h22ds203@hirosaki−u.ac.jp)

**Abstract**.

We investigate bidirectional wake effects between the same type of two wind turbines in a hill region in northern Japan using Supervisory Control and Data Acquisition (SCADA) data and validate the simulated wakes by 12 wake models. The extent to which complex terrain affects wake behavior has not yet been fully studied and further understanding of fundamental characteristics of bidirectional wakes over complex terrain is required. The two wind turbines are located 3.17 times the rotor diameter apart with a different elevation of 0.44 times the rotor diameter. First, we identify the wake effects in terms of wind speed ratio, which is defined as a ratio of wind speed at the downstream wind turbine to that at the upstream wind turbine. By comparing the conditions according to the operating state of the upstream wind turbine, the wakes are clearly detected as minimum wind speed ratios for northeasterly and southwesterly winds. The wind speed ratio with a maximum wake effect occurs below the rated wind speed. Increase in turbulence intensity and decrease in power output are greater for southwesterly wind than for northeasterly wind. This difference arises from the combined effects of the turbine-induced wake and the terrain-induced variation in wind speed. Then, we examine the winds and the wakes over complex terrain by using Wind Atlas Analysis and Application Program Computational Fluid Dynamics (WAsP CFD) in combination with PyWake. The wind speed ratios derived from the wake models show strong dependence on inflow wind speed, reflecting the thrust coefficient curve used in the wake models. The wake models commonly overestimate the reduction in wind speed ratio for northeasterly wind and underestimate it for southwesterly wind. This comparative study contributes to understanding the additional effects of topography on wake effects in onshore wind power plants and offshore wind power plants near the coast.

## 1 Introduction

Considering effects of wakes generated by neighbouring wind turbines is crucial to design a layout of wind power plants. We need to manage the wake effects, and otherwise, wakes can negatively impact on downstream wind turbines by reducing wind speeds and power output and by increasing turbulence intensity and load fluctuation. Moreover, the topography affects the wakes and makes the airflow more complex. An increasing number of wind turbines are being installed in complex terrains, such as hills, escarpments, valleys, and gorges, and flat terrain suitable for wind power plants. Although installation in complex terrain poses logistic challenges, strong winds induced by topographic effects offer an advantage for wind resource (e.g., Yamaguchi et al., 2024). Thus, the combined effects of wake and terrain further complicate the optimization of the performance and control strategies of wind power plants.

Analyzing wind measurement data is essential for understanding wake phenomena (e.g., Rhodes and Lundquist, 2013; Hansen et al., 2016; El-Asha et al., 2017; Mittelmeier et al., 2017; Astolfi et al., 2018; Han et al., 2018). Among various observational data, supervisory control and data acquisition (SCADA) provides the advantage of enabling concurrent analysis of wind conditions at hub height and operating states of wind turbines. In addition to the SCADA data, external measurements, such as light detection

and ranging (lidar) measurements, are widely used to observe wakes. Combined use of various observations enables to further investigate the impact of wakes on power generation of downstream wind turbines in flat terrain (El-Asha et al., 2017) and in complex terrain (Han et al., 2018), and on turbulence intensity at downstream wind turbines (Mittelmeier et al., 2017). Rhodes and Lundquist (2013) indicate dependence of wake effects on the inflow wind speed and the vertical structures of reduction in wind speed and of increase in turbulence intensity due to wakes by installing two Doppler lidars for upstream and downstream a wind turbine. Astolfi et al. (2018) use SCADA data with numerical simulation to demonstrate that the wake structures are distorted by the surrounding terrain and that wind speed recovery is quicker over complex terrain than over flat terrain. Using SCADA data in conjunction with external measurements is an effective method for wake detection. However, using external measurements involves high costs for operation and maintenance, and operation periods of the external measurements are often limited.

Numerical simulations enable to effectively analyse the wake and complement observations (Göçmen et al., 2016; Carbajo Fuertes et al., 2018; Bastankhah and Porté-Agel, 2014; Fischereit et al. 2022; Troldborg et al., 2022). Göçmen et al. (2016) show that wake models have different advantages in terms of usability, accuracy, quantity and quality of input data, and computational costs by evaluating utilization of the six wake models with SCADA data in onshore and offshore wind power plants. Gaussian wake model is validated by two pulsed scanning Doppler lidars mounted on the nacelle, which measure both the incoming flow and the downstream wake (Carbajo Fuertes et al., 2018). Bastankhah and Porté-Agel (2014) validate a Gaussian model for the reproducibility of wind speed at the downstream wind turbine by wind-tunnel measurements and the large-eddy simulations (LES). Fischereit et al. (2022) indicate that the selected wake models accurately represent intra-farm wakes; however, they underestimate the farm-farm wake effects compared with high-fidelity simulations. Troldborg et al. (2022) found that the performance of wind turbines in complex terrain is highly variable compared to that in flat homogeneous terrain by using LES. Wake models have been validated by various methods; however, a limited number of studies have validated simulated wakes over the complex terrain.

The following studies have investigated fundamental behaviour of wakes over complex terrain. The measurement campaign in a double-ridge site using the meteorological measurement in Perdigão field in 2017 is well-known observational field experiment over a complex terrain (Wagner et al., 2019). The wake generated by a single wind turbine at the ridge goes up because a strong recirculation zone is creased in a downstream (e.g., Berg et al., 2017; Dar et al., 2019; Wenz et al., 2022). Letzgus et al. (2022) demonstrate that high turbulence flow due to the topography results in asymmetric torque distributions and leads to increased load fluctuations of a wind turbine in a flat plateau downstream of escarpment by using numerical simulation. However, Porté-Agel et al. (2020) mention that the combined study of wind turbine wakes and topography has still much room to development. Simulation of wind over complex terrain is still a challenge (Castellani et al., 2015) and the inflow wind speed over complex terrain is unclear (Politis et al., 2011). Yang et al. (2015) show the decrease in wind speed and increase in the turbulence intensity induced by the upstream trench using several hypothetical wind turbines. Fleming et al. (2019, 2020) demonstrate a 14% increase in energy production by applying wake steering to an upstream wind turbine and a downstream wind turbine. Sasanuma and Honda (2022; 2024) focus on two onshore wind turbines in complex terrain in northern Japan (Fig. 1) and investigate the wake effects of one turbine on the other. The southwesterly wind results in a significant reduction in wind speed and an increase in turbulence intensity at the downstream wind turbine (Sasanuma and Honda, 2022). Sasanuma and Honda (2024) indicate that wake models generally underestimate the reduction in wind speed in the wakes compared with the observations. Further advancements in the study of the fundamental behavior of wakes over complex terrain are required to evaluate wake effects induced by multiple upstream wind turbines or by entire wind power plants.

A detailed investigation of the bidirectional wakes between two wind turbines is effective. First, we need to compare bidirectional wake effects between the two wind turbines to clarify the differences in wake effects over the complex terrain according to the wind directions. Sasanuma and Honda (2022; 2024) investigate wake effects only one-way direction between the two wind turbines. Wake effects in the other direction remain unclear. Moreover, we need to look into the dependence of wake effects on inflow wind speed over complex terrain. Second, we need to validate the performance of the wake models by comparing the simulated results with the observed results and clarifying the dependence of simulated wake effects on inflow wind speed. The validation of simulated

wakes over complex terrain is still insufficient. By comparing the mutual wake effects between wake models, we need to identify similarities and differences among the wake models.

We statistically investigate the bidirectional wakes behavior between two wind turbines over the complex terrain using the SCADA data and validate the performance of wake models in terms of reduction in wind speed at the downstream wind turbine by comparing with the SCADA data. We specifically address the following two issues: (1) to identify the wake effects at the downstream wind turbine by comparing the conditions with and without the operation of the upstream wind turbine and evaluate the wake effects in terms of reduction in wind speed, turbulence intensity, and power output. (2) to demonstrate the difference in reproducibility of mutual wakes between the two wind turbines simulated by 12 wake models.

Section 2 describes the wind turbines, the SCADA data, the wind condition at the site, the methodology to detect wakes, and the flow simulation with wake models. Section 3 analyses the SCADA data to examine the wake effects and compares the observed results with the simulated results by wake models. Section 4 presents the summary and conclusions.

## 2 Data and Methodology

### 2.1 Study site and two wind turbines

This study focuses on two wind turbines (WT1 and WT2) located near the northern tip of the Tsugaru Peninsula, Japan (Fig. 1). Because abundant wind resources and steep topography characterize the region near Cape Tappi, this site has been a research site for operation of wind turbines since the first utility-scale wind farm in Japan was built in 1991 (Fig. 1; e.g., Ushiyama, 1999; Inomata et al., 1999; Enomoto et al., 2001; Fujikawa et al., 2002; Hasegawa et al., 2003). The two wind turbines are originally J82-2.0 models manufactured by Japan Steel Works. To satisfy safety requirements for fatigue strength in this site, their hub height is modified from 80.0 m to 65.0 m, and the rated power output is reduced from 2 MW to 1.675 MW (Fig. 2a). The rotor diameter is 83.3 m. Full operation started in 2010. The two wind turbines are located almost in a southwest-northeast direction (Fig. 1) and the horizontal distance between the two turbines is 3.17 times the rotor diameter (264 m; Fig. 2b). WT1 and WT2 are located at an altitude of 132 m and 169 m, respectively. The difference in elevation correspond to 0.44 times the rotor diameter. Along the line connecting the two turbines, WT2 is located at the highest location (Fig. 2b). The surrounding terrain is covered by bush trees (Fig. 2a).

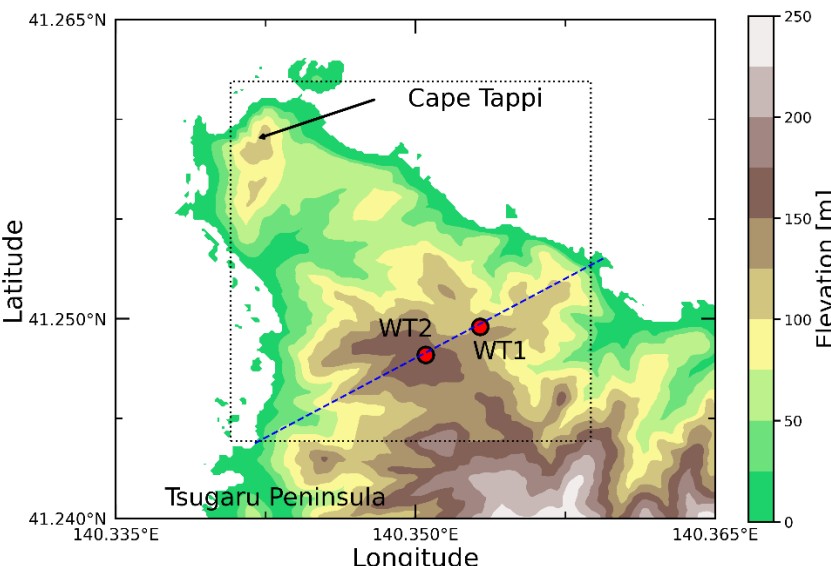

**Figure 1.** The topography of the study site (color shading) and the locations of the two wind turbines (WT1 and WT2; red circles). The blue dashed line represents the line connecting the two wind turbines to show the vertical topography in Fig. 2b and to analyze the vertical cross sections in Fig. 14. The square enclosed by the dotted line is a 2 km × 2 km computational domain for WAsP CFD.

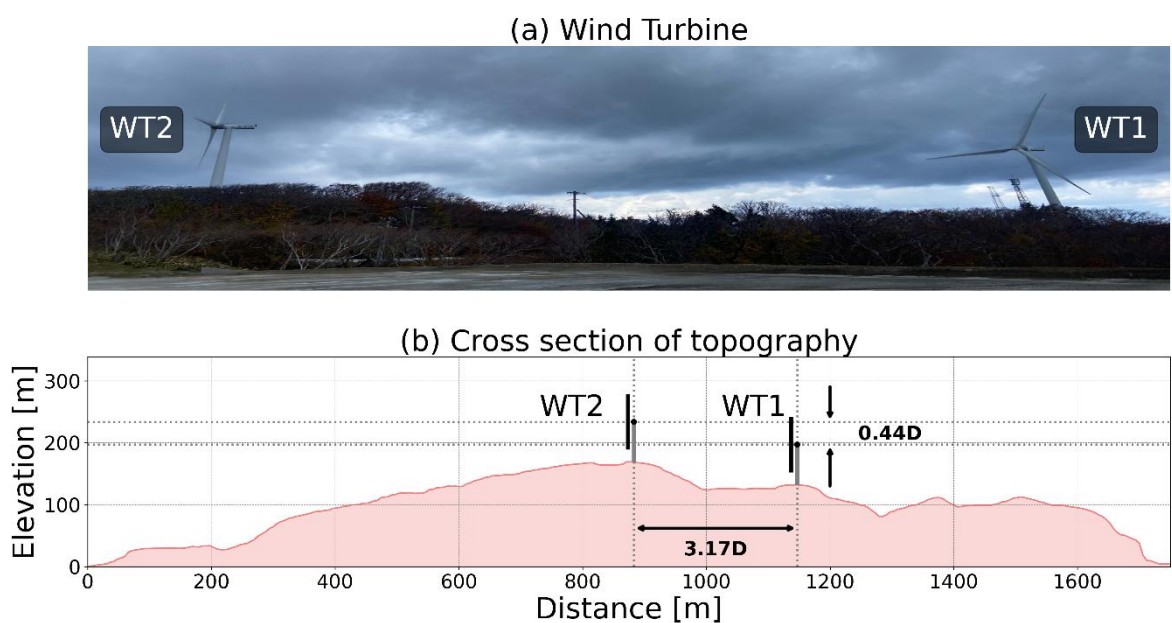

**Figure. 2** (a) Photo of the two wind turbines. (b) Cross section of the topography along the line connecting the two wind turbines shown in Fig. 1. The elevation data are obtained from the website of Geospatial Information Authority of Japan (Geospatial Information Authority of Japan, 2025).

**2.2 SCADA data at the two wind turbines**

We used 10-minute mean SCADA data of the two wind turbines from 5 September 2015 to 31 December 2017. The original data are sampled at an interval of 1 second. The dataset includes wind speed, wind direction, standard deviation of wind speed, nacelle direction, blade pitch angle, rotor speed, and power output. The wind data are measured by a wind vane and a three-cup type

anemometer mounted at the rear of the nacelle. Turbulence intensity is defined as $\sigma$ /WS, where $\sigma$ is standard deviation of wind speed, and WS is the 10-minute mean wind speed. To effectively investigate the dependency of the wind turbine wake on wind direction, we used the data with the yaw misalignment, a difference between wind direction and nacelle direction, within plus or minus 10°. A total of 80% of the data satisfy this condition. In the following analyses, we used the SCADA data after applying this selection based on yaw misalignment.

## 2.3 Wind climate

We present the wind climate at the two wind turbines from the SCADA data (Figs. 3 and 4). The westerly wind and easterly wind commonly dominate at WT1 and WT2 (Fig. 3). Strong westerly winds dominate in winter due to the Siberian high, whereas easterly winds intermittently blow in summer due to the Okhotsk high (Shimada et al., 2014). Westerly winds are stronger than easterly wind and the wind direction from south-southwest to west-northwest accounts for 70% at WT2 (Fig. 3a) and 69% at WT1 (Fig. 3b). Both wind turbines have abundant wind resources with annual mean wind speed above 8 m s$^{-1}$. WT2 exhibits higher speeds than WT1 because WT2 is located on top of the hill, and the wind is accelerated on the hill (Fig. 2b; Yamaguchi et al., 2002). The frequency exceeding 15 m s$^{-1}$ amounts to 5% at WT1 and 14% at WT2 in all wind directions. The overall results are consistent with the results without data selection based on yaw misalignment (Sasanuma and Honda, 2022, 2024). The wind speeds at WT1 and WT2 show significant correlation (Fig. 4a). Wind speeds at WT2 are 16% higher than those at WT1. Wind directions at WT1 and WT2 are in good agreement (Fig. 4b). The data are concentrated in the easterly wind (90°–100°) and the westerly wind (220°–260°), which is consistent with Fig. 3. The wind blowing from WT1 to WT2 account for 2% and the wind blowing from WT2 to WT1 account for 17%. At these frequencies, wakes are expected to occur. Thus, the locations of the two wind turbines are suitable for analysing mutual wake effects between the wind turbines.

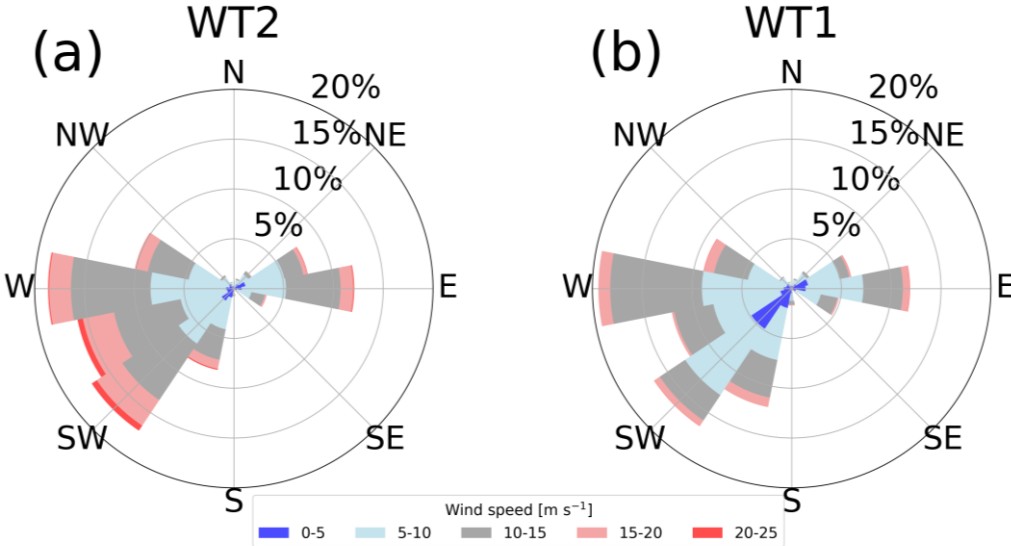

**Figure 3.** Wind roses derived from the SCADA data of (a) WT2 and (b) WT1. The different colors are used at every 5 m s$^{-1}$. Wind direction is defined as the direction from which the wind blows.

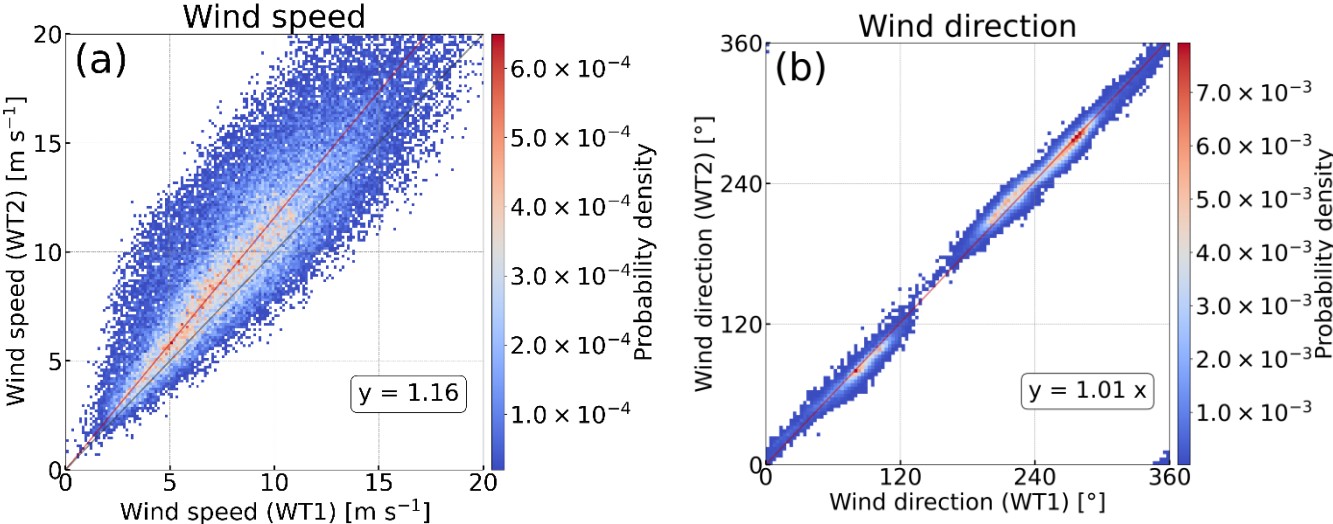

**Figure 4.** Scatter plots of (a) wind speed and (b) wind direction between WT1 and WT2 from the SCADA data. The color represents the data density. The red lines in (a) and (b) are the linear regression lines, shown by the equations in the figures. The black line in (a) represents a 1:1 line.

## 2.4 Detection of wake effects

To assess the wake effects generated by the upstream wind turbine based on the SCADA data, we divided the data into two categories based on the operating state of the upstream wind turbine: "no-wake conditions" and "wake conditions" (Sasanuma and Honda, 2022; Sasanuma and Honda, 2024). "No-wake conditions" mean that the upstream wind turbine is not operating and "wake conditions" mean that the upstream wind turbine is in operation. Sasanuma and Honda (2024) identify the operating states of the upstream wind turbines by a combination of rotor speed, blade pitch angle, and power output. We found that we can distinguish whether the wind turbine is operating or not only by the blade pitch angle. By imposing the condition that the blade pitch angle is within ±3°, we can extract data when the upstream wind turbine is in operation below the rated wind speed. The blade pitch angle is 90° when the wind turbine is not operating. When considering wake conditions, we focus on the data with wind speeds ranging from the cut-in wind speed to the rated wind speed. In this range, wake effects are maximized; however, above the rated wind speed, wake effects are reduced by the blade pitch control (e.g., El-Asha et al., 2017; Dilip and Porté-Agel, 2017; Duncan et al., 2020).

Thus, by comparing no-wake conditions and wake conditions, we can statistically analyse the influence of wakes generated by the upstream wind turbine on wind speed, turbulence intensity, and power output at the downstream wind turbine.

## 2.5 Wind speed ratios

To analyse mutual wake effects between the two wind turbines, we define a wind speed ratio, which is a ratio of the wind speed at the downstream wind turbine to that at the upstream wind turbine. We should notice that the definition of the wind speed ratio between wind speed at WT1 ($WS_1$) and that at WT2 ($WS_2$) differs by wind directions. When northeasterly wind blows, the wake induced by WT1 influences WT2. Thus, the wind speed ratio is defined as $WS_2 / WS_1$. When southwesterly wind blows, the wake induced by WT2 influences WT1. Thus, the wind speed ratio is defined as $WS_1 / WS_2$. We summarize the wind speed ratios in no-wake conditions and in wake conditions for the respective northeasterly wind and southwesterly wind in Table 1. We compare the wind speed ratio between no-wake and wake conditions for northeasterly wind and southwesterly wind to evaluate the wake effects in the following analyses.

**Table 1.** Definition of wind speed ratio according to the wind directions and the presence or absence of wakes. White shading indicates the condition that the upstream wind turbine is not in operation, and gray shading indicates the condition that the upstream wind turbine is in operation.

|  | Northeasterly wind | Southeasterly wind |
|---|---|---|
| No-wake conditions | $WS_2 / WS_1$ | $WS_1 / WS_2$ |
| Wake conditions | $WS_2 / WS_1$ | $WS_1 / WS_2$ |

## 2.6 Flow simulations with wake models

We utilized Wind Atlas Analysis and Application Program Computational Fluid Dynamics (WAsP CFD; Bechmann, 2012) in combination with PyWake (Pedersen et al., 2023) to closely examine the observed wind and wakes over complex terrain. WAsP CFD is a CFD model integrated into WAsP and is designed for simulating winds over complex terrain. The model is based on Ellipsys 3D code, which is a multiblock finite volume discretization of incompressible Reynolds Averaged Navier-Stokes (RANS) equations. The two-equation k-ε RANS model by Lauder and Spalding (1974) is used for turbulence. Simulations are conducted under the assumption of a neutral atmospheric stability. Cariolis forcing is neglected. The validity of WAsP CFD in complex terrain has been demonstrated (e.g., Bechmann et al., 2011; Troen et al., 2015; Bechmann, 2016; Sharma et al., 2020). We used WAsP CFD for evaluating the influence of topography on wind flow. Computational domain is shown in Fig. 1. The horizontal calculations were conducted at 90 m intervals, and the elevation map is spaced at 5 m intervals. (Geospatial Information Authority of Japan, 2025). The output data are at 13 vertical levels from 5 m to 300 m (5, 10, 20, 33, 48, 65, 80, 100, 120, 150, 200, 250, and 300 m) in every 5° bin of wind direction. The upper boundary of the computational domain is 7000 m. We set the inflow wind speed far upwind to 10 m s$^{-1}$ at 100 m height, and the roughness length is assumed to be 0.0032 m, corresponding to the sea surface. Therefore, the inflow wind speed has a slightly sheared profile. WAsP CFD uses the roughness length depending on the landscape type. The roughness length in the study area ranges from 0 to 1.5 m.

PyWake is an open-source wind farm simulation tool written in Python. This tool simulates the flow with wind turbines by incorporating wake models. PyWake has 12 wake models (Appendix A): 11 engineering wake models and a linearized RANS wake

model (Fuga). Note that we use the default parameter values of the 12 wake models without adjusting the parameters to obtain the results close to the SCADA data. The sensitivity analysis of the model parameters is beyond the scope of this study. Bastankhah model and TurboGaussian model, which we particularly focus on, have a reduction in wind speed with a Gaussian profile (Bastankhah and Porté-Agel, 2014; Pedersen et al., 2022). They have been validated for onshore field (e.g., Jeon et al., 2015; Ruisi et al., 2019; Fleming et al., 2020; zum Berge et al., 2024) and offshore filed (Farrell et al., 2021; Fischereit et al., 2022; Centurelli et al., 2024). The turbulence intensity is set to 0.2 in PyWake. We used the thrust coefficient curve of the V80 2 MW wind turbine because the rated power output of this wind turbine is closest to that of the wind turbines in this study among the wind turbine models available in PyWake. The thrust coefficient curve of the V80 is shown in Fig. 5. In this study, we discuss similarities and differences between the wake models for mutual wind directions when a typical thrust coefficient curve is used.

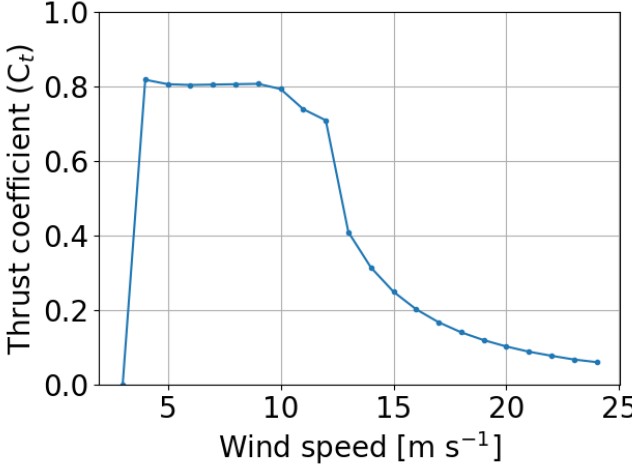

**Figure 5.** Thrust coefficient as a function of wind speed from V80 wind turbine.

# 3 Results

## 3.1 Observational results

We examine the wind speed ratios to evaluate wake effects caused by the upstream wind turbine. Figure 6 shows wind speed ratios between the two wind turbines before we divided the data into two categories on the basis of the operating state of the upstream wind turbine. The wind speed ratio varies in wind direction due to the surrounding terrain and wakes. The wind speed ratios are generally above 1.0 in Fig. 6a and below 1.0 in Fig. 6b, whereas we can identify V-shaped curves or significant reductions in wind speed ratio at around 40° and 225° (light blue shading in Figs. 6a and 6b). These wind directions correspond approximately to the direction of the line connecting the two wind turbines (Fig. 1). The ranges of these significant reductions in wind speed ratio are approximately ±10° on both sides of the wind directions. We can see no other significant reduction in the remaining wind direction ranges. We focus on these two wind directions in the following analyses.

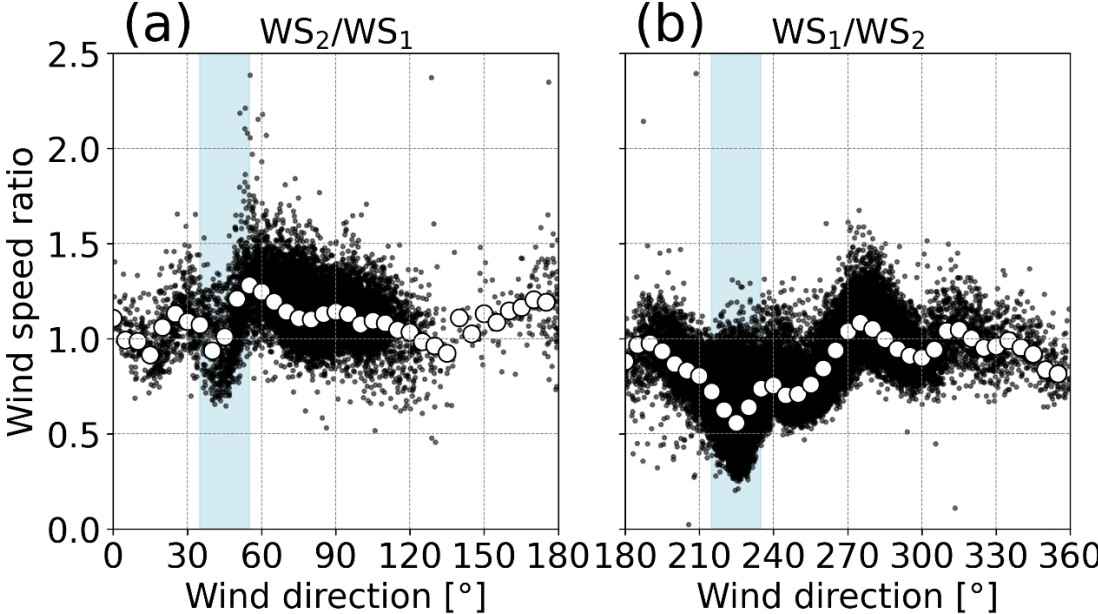

**Figure 6.** Wind speed ratios as a function of wind direction. Black dots denote 10-minute mean data and white circles denote median values in every 5º bin. In (a) and (b), wind directions at WT2 are commonly used for horizontal axes. Note that the definition of the wind speed ratio differs between (a) and (b). In (a), the wind speed ratio is defined as the ratio of wind speed at WT2 ($WS_2$) to that at WT1 ($WS_1$). In (b), the wind speed ratio is defined as the ratio of $WS_1$ to $WS_2$. These definitions of the wind speed ratio are common in the following analyses. Light blue shading indicates the ranges of wind direction with wake effects for (a) 45º±10º and (b) 225º±10º.

To clearly elucidate wake effects from the results of Fig. 6, we divide the data by the operating state of the upstream wind turbine. The condition of blade pitch angle is applied, and resultantly the data with wind speeds higher than the rated wind speed is excluded. We compare the wind speed ratio between no-wake conditions and wake conditions with a focus on the V-shaped curves at around 45°±10° and 225°±10° (Fig. 7). For northeasterly wind, the plots show no V-shaped curve and remain generally constant in no-wake conditions (Fig. 7a). The wind speed ratios are higher than 1.0 with a median value of 1.18 at 45° and this means that the wind speeds at downstream WT2 are higher than those at upstream WT1. In wake conditions, we can see a significant reduction in wind speed ratio due to the wake of WT1 with a maximum reduction of 23% at wind direction of 45° (Fig. 7b). For southwesterly wind, as is the case with Fig. 7a, the plots show no V-shaped curve and almost the constant wind speed ratio in no-wake conditions (Fig. 7c). The median value of wind speed ratio is 0.87 at 225°. The wind decelerates from the location of WT2 to the location of WT1 even without wakes generated by WT2 because WT1 is located behind the hill. In wake conditions, wind speed ratio decreases to 0.47 from 0.87 in no-wake conditions at 225°, and this means a 46% reduction (Fig. 7d). This reduction observed for southwesterly wind is greater than that observed for northeasterly winds. This reduction is 22% larger than that derived by Sasanuma and Honda (2022) because they use the mean values rather than the median values. From the above results, we find that the operation of the upstream wind turbine causes wind speed reduction at the downstream wind turbine and that the wake effect on the downhill side for southwesterly wind is more pronounced than that on the uphill side for northeasterly wind.

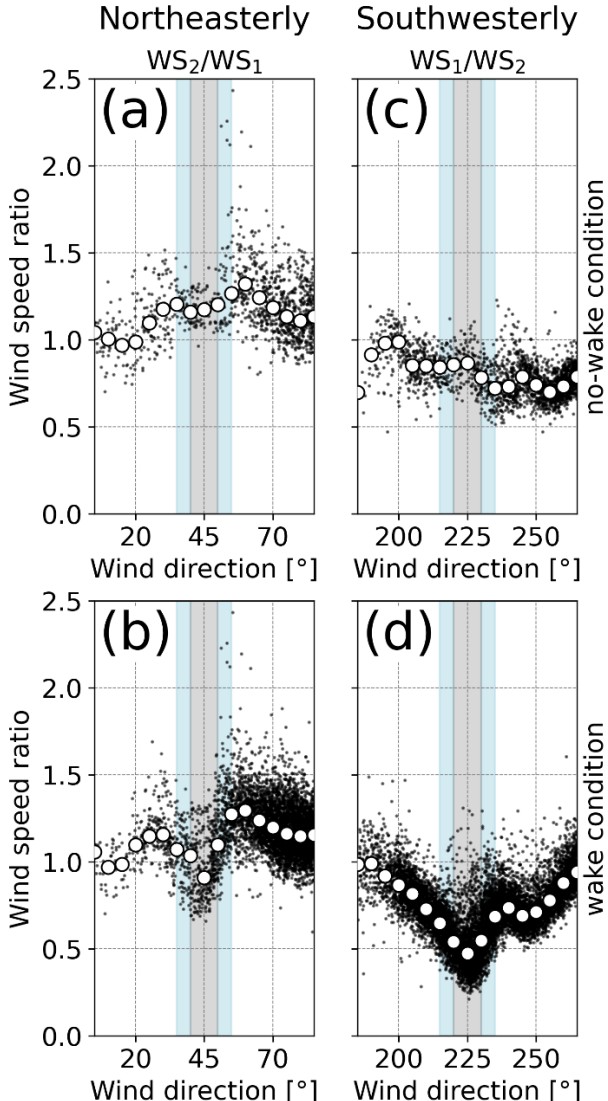

**Figure 7.** The same as in Fig. 6, but for being classified by the operating state of the upstream wind turbine in (a) (c) no-wake
conditions and (b) (d) wake conditions for (a) (b) northeasterly wind and (c) (d) southwesterly wind. Light blue and gray shadings
indicate the range of wind direction with wake effects. The gray shading in the ranges of (a) (c) $45°\pm5°$ and (b) (d) $225°\pm5°$ represent
the focused ranges of wind direction for the analyses in Figs. 8, 9, and 10.

We examine the dependence of the magnitude of reduction in wind speed on inflow wind speed in wake conditions (Fig. 8). To
examine the maximum wake effects, we focus on the ranges of wind direction for $45°\pm5°$ and $225°\pm5°$ (gray shading in Figs. 7b
and 7d). We use the wind speed of the upstream wind turbine in operation as the inflow wind speed on the horizontal axis. Figure 8
clearly shows the relationship between the inflow wind speed and wind speed ratio for both northeasterly and southwesterly winds.
The decrease in wind speed ratio means the increase in wake effect or the reduction in wind speed at the downstream wind turbine.
For northeasterly wind, the wind speed ratio decreases as the inflow wind speed increases below a wind speed of 6 m s$^{-1}$, whereas
above 6 m s$^{-1}$, the wind speed ratio remains almost constant (Fig. 8a). For southwesterly wind, the wind speed ratio decreases with
inflow wind speed and reach a minimum at 10 m s$^{-1}$ (Fig. 8b). The gradual increase for speeds above 10 m s$^{-1}$ suggests a start of
control of the blade pitch angle. Thus, the wind turbine significantly extracts the kinetic energy from the wind below the rated wind
speed, and the resulting momentum loss in the downstream induces the maximum wake effects. These results are consistent with
those of Rhodes and Lundquist (2013), showing that the maximum reduction in wind speed occurs below the rated wind speed.

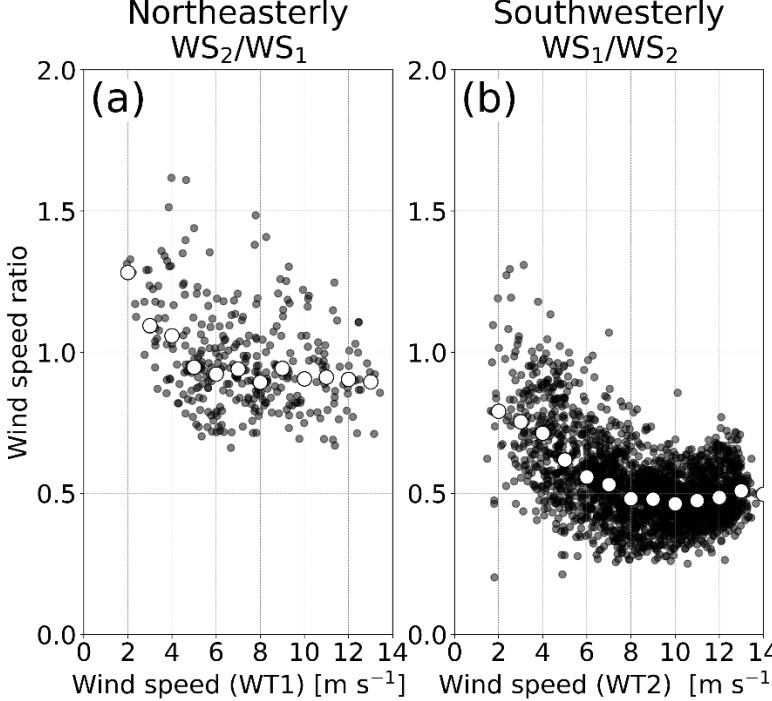

**Figure 8.** Wind speed ratio as a function of wind speed at the upstream wind turbine in operation for (a) northeasterly wind and (b) southwesterly wind. Black dots denote 10-minute mean data and white circles denote median values in every 1 m s$^{-1}$ bin. The data are extracted from the focused range of wind direction, denoted by the gray shading in Figs. 7b and 7d.

We investigate the wake effects on the turbulence intensity at the downstream wind turbine by comparing the observed turbulence intensity with the standard curves of turbulence intensity defined by the International Electrotechnical Commission (IEC, 2019). Two categories of the standard curves are given by the following equations. WS is wind speed.

$$\text{Turbulence instensity}_{\text{category}}A \ = \ 0.16\,(0.75 \ + \ 5.6\,/\,WS) \tag{1}$$

$$\text{Turbulence instensity}_{\text{category}}A^+ \ = \ 0.18\,(0.75 \ + \ 5.6\,/\,WS) \tag{2}$$

For northeasterly wind, the 90th percentile values of turbulence intensity hardly exceed the line of A+ category commonly in no-wake conditions and wake conditions (Figs. 9a and 9b). The wakes generated by WT1 contribute little to the increase in turbulence intensity at WT2. For southwesterly wind, the 90th percentile values of turbulence intensity for wind speeds greater than 7 m s$^{-1}$ follow the line of the A+ category in no-wake conditions (Fig. 9c; Sasanuma and Honda, 2022). In wake conditions, wind speed at WT1 decreases to less than 10 m s$^{-1}$ due to the wake generated by WT2, and the 90th percentile values of turbulence intensity follow the lines of A+ category and A category (Fig. 9d). However, the 10-minute mean data of turbulence intensity above the 90th percentile significantly exceeds the line of A+ category. The 10-minute mean data of turbulence intensity exceed 0.3 for wind speed less than 6 m s$^{-1}$. This is a noticeable difference between no-wake conditions and wake conditions. These results suggest that the wake effects on the turbulence intensity significantly vary depending on the wind directions.

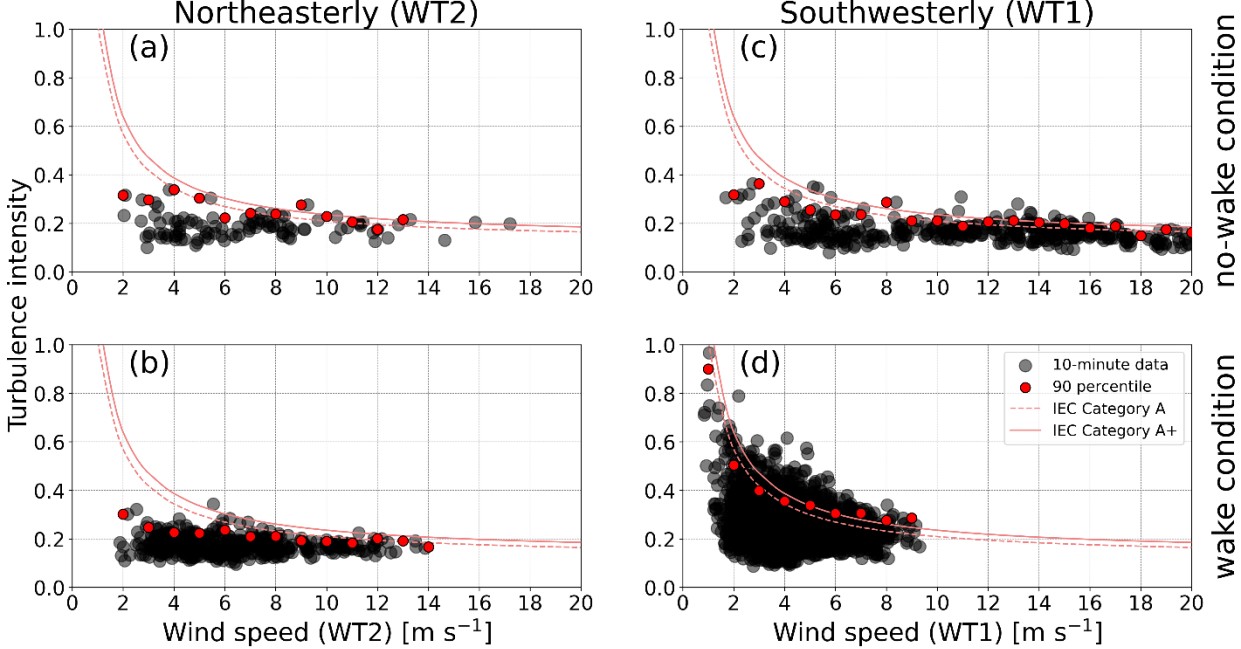

**Figure 9.** Turbulence intensity as a function of wind speed at the downstream wind turbine in (a) (c) no-wake conditions and (b) (d) wake conditions for (a) (b) northeasterly wind and (c) (d) southwesterly wind. The horizontal axis represents wind speed at the downstream wind turbine, $WS_2$ in (a) and (b), and $WS_1$ in (c) and (d). Black dots denote 10-minute mean data and red circles denote 90th percentile values in every 1 m s$^{-1}$ bin. The red dashed lines indicate standard curves of International Electrotechnical Commission (IEC) category A for high turbulence by equation (1), and the red solid lines indicate standard curves of IEC category A+ for very high turbulence by equation (2). The data are extracted from the focused range of wind direction, denoted by the gray shading in Fig. 7.

The impact of the wake on the power output of the downstream wind turbine is examined within the ranges of wind direction of 45°±5° and 225°±5° (Fig. 10). Note that the horizontal axis in Fig. 10 represents wind speed at the upstream wind turbine to examine the reduced power output at the downstream wind turbine compared with the power output at the upstream wind turbine. For northeasterly wind, the power output at the downstream wind turbine WT2 is lower than that expected from the wind speed at the upstream wind turbine WT1 (Figs. 10a and 10b). For example, in no-wake conditions, the normalized power output is approximately 0.5 at a wind speed of 8 m s$^{-1}$ (Fig. 10a). In wake conditions, a wind speed of 9 m s$^{-1}$ or a 12.5% increase in wind speed at the downstream wind turbine is required to generate the same power output (Fig. 10b). For southwesterly wind, the power output at the downstream wind turbine WT1 decreases more significantly than that expected from the wind speed at the upstream wind turbine WT2 (Figs. 10c and 10d). For instance, the normalized power output is approximately 0.2 at 7 m s$^{-1}$ in no-wake conditions (Fig. 10c). In wake conditions, a wind speed of 12 m s$^{-1}$ or a 71% increase in wind speed is required to generate the same power output (Fig. 10d). At 14 m s$^{-1}$, the normalized power output in no-wake conditions is approximately 0.9 (Fig. 10c), whereas it decreases to approximately 0.3 in wake conditions (Fig. 10d). This result represents a 67% reduction in power output in case that we assume the same inflow wind speed at the downstream wind turbine with that at the upstream wind turbine. These findings suggest that the wake effects have a more pronounced impact on power output for southwesterly wind than northeasterly wind. Even when the same wind speed is observed at the upstream wind turbine for northeasterly wind and southwesterly wind, the degree of reduction in wind speed or the resulting power output at the downstream wind turbine is different depending on the installation conditions of wind turbines.

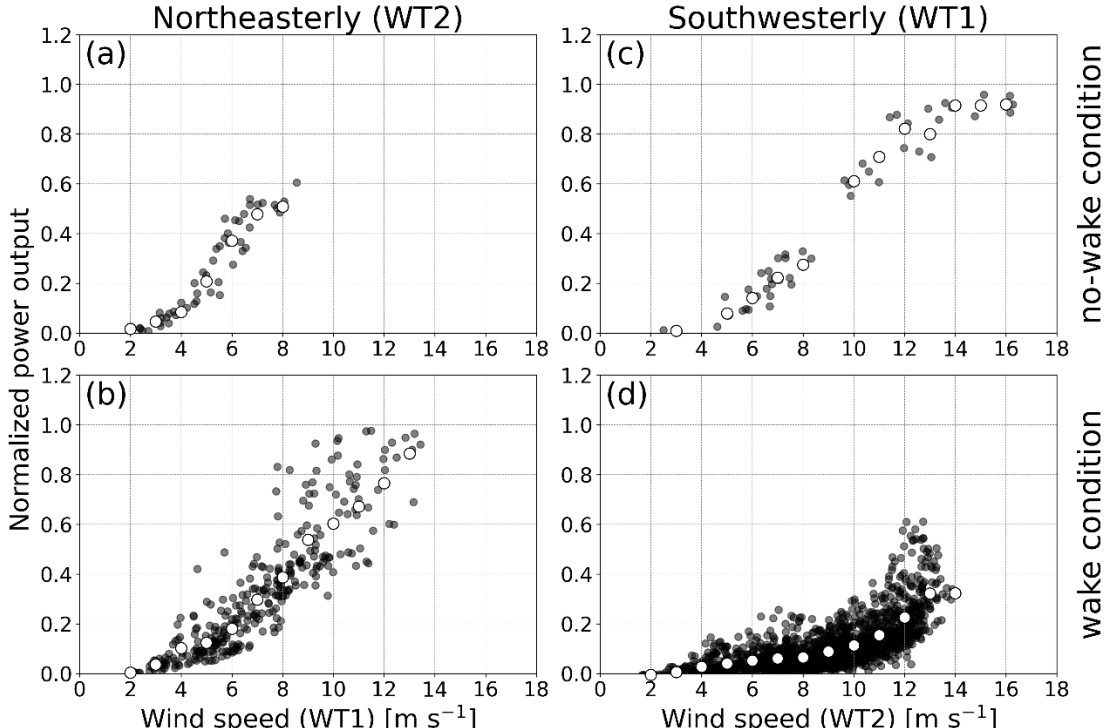

**Figure 10.** Normalized power output as a function of wind speed at the upstream wind turbine in (a) (c) no-wake conditions and (b) (d) wake conditions for (a) (b) northeasterly wind and (c) (d) southwesterly wind. The normalized power output in the vertical axis represents power output divided by the rated power output. The horizontal axis represents wind speed at the upstream wind turbine, $WS_2$ in (a) and (b), and $WS_1$ in (c) and (d). Black dots denote 10-minute mean data and white circles denote median values in every 1 m s$^{-1}$ bin. The data are extracted from the focused range of wind direction, denoted by the gray shading in Fig. 7.

## 3.2 Simulated wake and wind flow

We validate the performance of 12 wake models in PyWake by comparing the wind speed ratios computed from the wake models with those from the SCADA data. We show the comparison results of wind speed ratio for different inflow wind speeds computed by WAsP CFD and a typical wake model, the TurboGaussian wake model for northeasterly and southwesterly winds (Fig. 11). The wind speed ratio decreases significantly with a decrease in the inflow wind speed in the ranges of wind direction with wake effects for both northeasterly winds and southwesterly winds. This means that a significant reduction in wind speed occurs when the inflow wind speed is small. The magnitudes of reduction in wind speed ratio are the same for wind speeds below 9 m s$^{-1}$ for northeasterly wind (Fig. 11a) and below 8 m s$^{-1}$ for southwesterly wind (Fig. 11b), resulting from the thrust coefficient curve used for the wake model (Fig. 5). The wake model shows the results close to the SCADA data at inflow wind speed of 13 m s$^{-1}$ for northeasterly wind and at inflow wind speed less than 8 m s$^{-1}$ for southwesterly wind. The model results overestimate the reduction in wind speed for northeasterly wind and underestimate it for southwesterly wind, which is generally consistent with the results of other wake models (Figs. B1 and B2). We can conclude that the topographic effects cause an opposite change in wake reproduction between northeasterly and southwesterly winds.

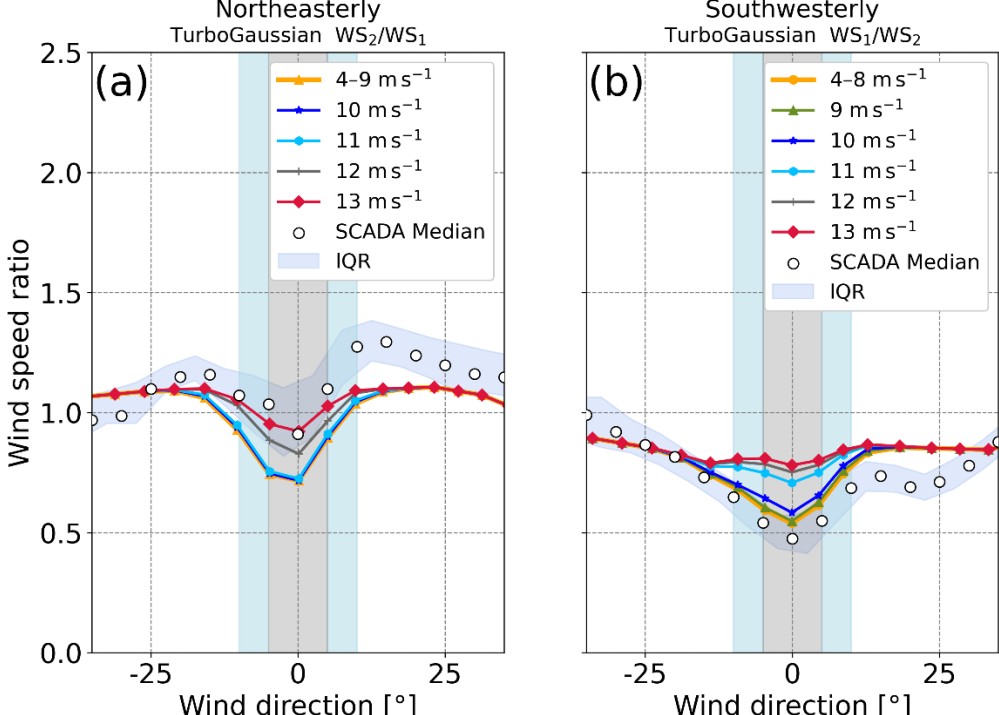

**Figure 11.** Wind speed ratio as a function of wind direction for (a) northeasterly wind and (b) southwesterly wind. Colored lines are results derived from the TurboGaussian wake model for inflow wind speed ranging from 4 to 13 m s$^{-1}$ at an interval of 1 m s$^{-1}$. The results for (a) 4–9 m s$^{-1}$ and (b) 4–8 m s$^{-1}$ are shown by one line because the differences are negligible. The horizontal axis indicates differences in wind direction, with the minimum wind speed ratio at 0º for both the SCADA data and the results of the wake model. The blue band denotes the interquartile range (IQR) of the SCADA data, and white circles denote median values in every 5º bin. Light blue and gray shadings indicate the ranges of wind direction with wake effects.

We compare the wind speed ratios derived from the 12 wake models at the inflow wind speeds at 13 m s$^{-1}$ for northeasterly wind and 8 m s$^{-1}$ for southwesterly wind (Fig. 12). For both northeasterly wind and southwesterly wind, most of the wake models underestimate the reduction in wind speed and represent weak wakes (Figs. 12a and 12b). For northeasterly wind, most of the wake models hardly represent the wakes (Fig. 12a), whereas for southwesterly wind, all the wake models except the GCL wake model represent a reduction of wind speed ratio (Fig. 12b). Only the Bastankhah and the TurboGaussian wake models closely represent the minimum value of wind speed ratio derived from the SCADA data for both northeasterly wind and southwesterly wind. To summarize the validation results from each wake model, we show the wind speed ratio for 12 wake models at wind directions with a maximum reduction in wind speed for both northeasterly and southwesterly winds in Fig. 13. The order of the wake models according to the difference of the results between each wake model and the SCADA data are almost the same for northeasterly and southwesterly winds (Figs. 13a and 13b). The results from the TurboGaussian and the Bastankhah wake models are consistent with those from the SCADA data, and the Blondel2020 and the Blondel2023 wake models follow the two wake models commonly for northeasterly and southwesterly winds. The differences from the SCADA data range from 0.001 to 0.151 for northeasterly wind and from 0.032 to 0.323 for southwesterly wind. The differences among the wake models are more prominent for southwesterly wind than for northeasterly wind.

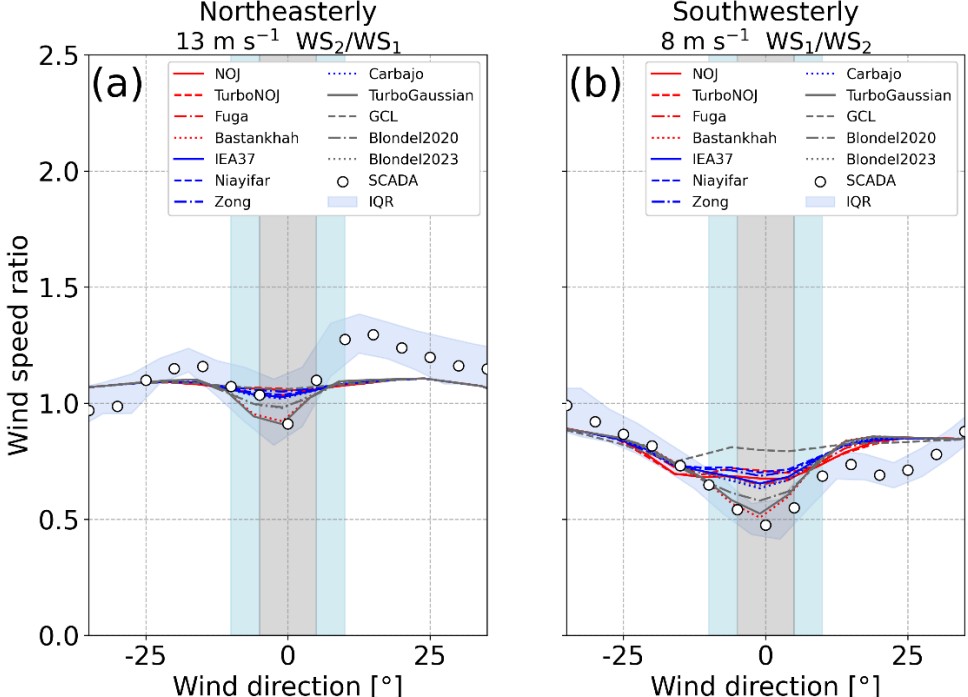

**Figure 12.** Wind speed ratio as a function of wind direction for (a) northeasterly wind and (b) southwesterly wind. Colored lines are results derived from 12 wake models at an inflow wind speed of (a) 13 m s⁻¹ and (b) 8 m s⁻¹. The horizontal axis indicates differences in wind direction, with the minimum wind speed ratio at 0º for both the SCADA data and the results of the wake models. The blue band denotes the interquartile range (IQR) of the SCADA data, and white circles denote median values in every 5º bin. Light blue and gray shadings indicate the ranges of wind direction with wake effects.

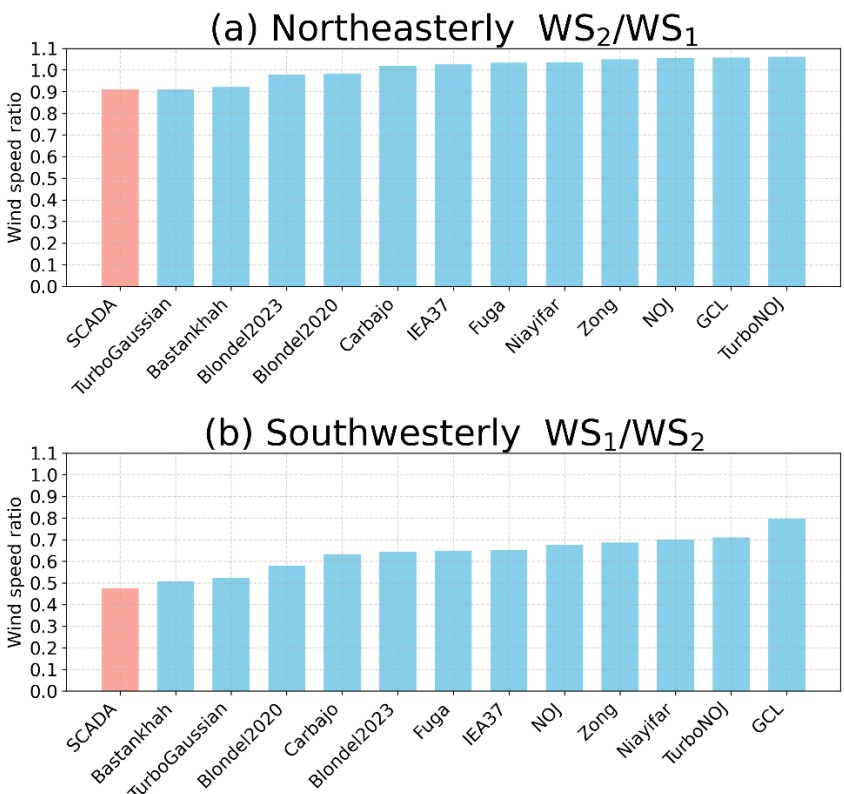

**Figure 13.** Wind speed ratio derived from the SCADA data and 12 wake models at wind direction with maximum wake effects at wind speed of (a) 13 m s$^{-1}$ for northeasterly wind and of (b) 8 m s$^{-1}$ for southwesterly wind.

To investigate the different wake effects between northeasterly and southwesterly winds, we show vertical structures of wind computed by WAsP CFD and PyWake along the line connecting the two wind turbines (Fig. 14). We computed wind flows with and without the wake effects with a far-upstream wind speed of 10 m s$^{-1}$ and derived the reduction in horizontal wind speed due to the wakes. In Figs. 14b and 14d, the TurboGaussian wake model is used. We can consider the path of the wakes generated by the upstream wind turbine from the streamline passing through the center of the wind turbine wakes (Sesarego et al., 2020). For

northeasterly wind, the wind reaches the hub height of WT2 from the hub height of WT1 due to the wind ascending the hill (Fig. 14a). Moreover, the flow accelerates over the top of the hill around the location of WT2, and the accelerated wind blows through the rotor surface of WT2. When the wind turbines are installed and operating, the Gaussian-shaped wake generated by WT1 rises upward along the terrain and reaches the center of the rotor of WT2 (Fig. 14b). However, the wake weakens at WT2 because the flow accelerates over the top of the hill. Thus, the wind speed ratios are generally higher than 1.0 due to the acceleration of wind

speed over the hill in no-wake conditions (Fig. 7a), and the wind speed ratios slightly decrease because the wake is weakened by the flow acceleration in wake conditions (Fig. 7b). For southwesterly wind, the weak winds in the lee of the hill of WT2 cover the location of WT1 and extend through the height of the lower part of the WT1 rotor (Fig. 14c). In contrast, the strong winds over the top of the hill reach the hub height of WT1. The resulting strong vertical wind shear at the height of the rotor of WT1 contributes to the increase in turbulence. When the wind turbines are installed and operating, the wake generated by WT2 extends horizontally

through the upper part of the rotor of WT1 (Fig. 14d). The wake does not follow the terrain slope in the lee of the hill, which is consistent with the previous studies (Berg et al., 2017; Dar et al., 2019; Wenz et al., 2022). Thus, the wind speed ratios in no-wake conditions are generally lower than 1.0 due to the acceleration of wind over the hill and the weak winds in the lee of the hill (Fig. 7c). The wind speed ratios decrease significantly in wake conditions due to the inclusion of the wake extending from the upstream wind turbine. (Fig. 7d).

From the results obtained so far, we consider the differences among the 12 wake models. The degree of wake reproducibility depends on inflow wind speed (Figs. 11, B1, and B2). Terrain effects induce a systematic bias in the reproducibility of wakes (Figs. 12, 13, B1, and B2). From the perspective of wake structure, the differences among the 12 wake models lie in the downstream extents of the wakes and the magnitude of the wind speed reduction immediately behind the wind turbine (Figs. 14, C1 and C2). In particular, the downstream extents of the wakes are key for obtaining results consistent with the SCADA observations. Although

the simulation results might depend on parameter settings of the wake models, the present results suggest that the major differences arise from the model formulation. These results highlight the importance of selecting appropriate wake models and considering topographic conditions.

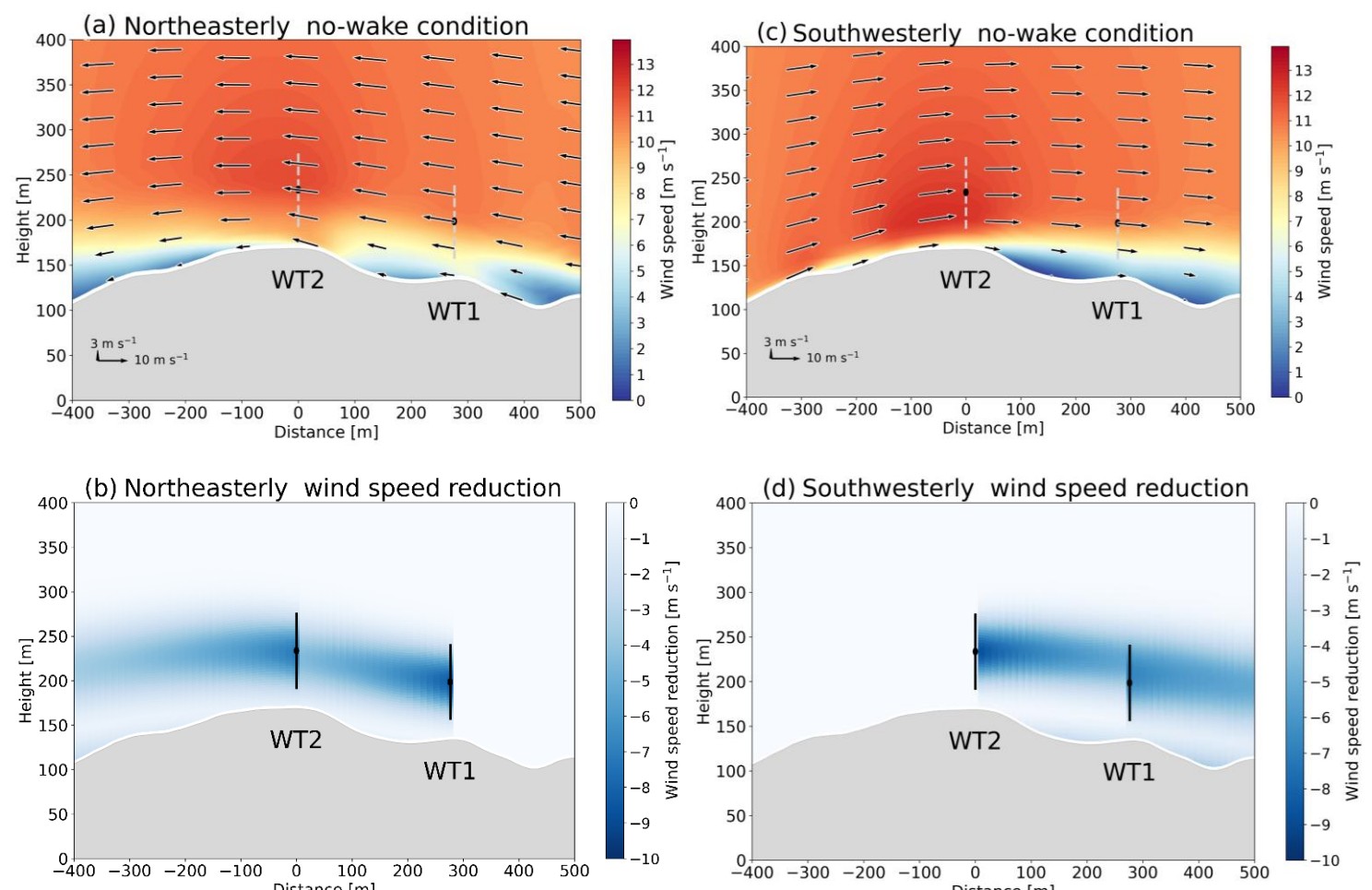

**Figure 14.** Vertical cross sections along the blue dashed line in Fig. 1 for (a) (b) northeasterly wind and (c) (d) southwesterly wind with a far-upstream wind speed of 10 m s⁻¹. In (a) and (c), we show horizontal wind speed (color shading) and wind vectors on the cross section with no wake effects of the wind turbines, although we show the positions of the wind turbine rotors by a gray dashed line. In (b) and (d), we show a reduction in horizontal wind speed (color shading) due to the wind turbine wake simulated by the

415 TurboGaussian wake model. A reduction in horizontal wind speed is the difference between no-wake conditions and wake conditions. The black lines indicate the positions of the wind turbine rotors. The data from WAsP CFD are available from 5 m above ground level.

## 4 Summary and conclusions

We have investigated the bidirectional wake effects between the two wind turbines in complex terrain at a site in northern Japan using the SCADA data and validated the performance of the wake models. We have addressed the two issues: (1) to identify the wake effects generated by the upstream wind turbine for northeasterly and southwesterly winds in terms of wind speed ratio, turbulence intensity, and power output at the downstream wind turbine. (2) to validate the performance of the 12 wake models and impacts of the topography on the simulated wakes. The main results are summarized as follows.

(1) Using the SCADA data, we detected wake effects from the differences in the wind speed ratio between no-wake conditions and wake conditions. These conditions are classified by the operating state of the upstream wind turbine based on the blade pitch angle. The upstream wind turbine in operating state reduces wind speed at the downstream wind turbine for both northeasterly and southwesterly winds. For northeasterly wind, the wind speed ratio in no-wake conditions generally exceeds 1.0 and the maximum

reduction in wind speed ratio due to the wake is 23%. For southwesterly wind, the wind speed ratio in no-wake conditions is generally below 1.0 and the maximum reduction in wind speed ratio due to the wake is 46%. Wake effects are enhanced more for southwesterly wind than for northeasterly wind. Wake effects reach their maximum below the rated wind speed for both northeasterly and southwesterly winds. Turbulence intensity at the downstream wind turbine increases due to the wake generated by the upstream wind turbine for southwesterly wind, and the 10-minute data of the turbulence intensity exceed the level of A+ category. The power output at the downstream wind turbine for southwesterly wind decreases more significantly than for northeasterly wind. From the analysis of the SCADA data, this study demonstrated that wake effects depend on the surrounding terrain and inflow wind speed. These results indicate the importance of considering the combined effects of wakes and topography.

(2) We compared the wind speed ratios derived from 12 wake models in PyWake with those from the SCADA data. The wind speed ratios derived from the wake models show strong dependence on the inflow wind speed. This dependence reflects the thrust coefficient curve used for wake computation in the wake models. The wake models commonly overestimate the reduction in wind speed ratio for northeasterly wind and underestimate it for southwesterly wind. The wake models with high reproducibility are the TurboGaussian and the Bastankhah wake models. These two wake models represent the minimum wind speed ratio or maximum reduction in wind speed close to the observations when inflow wind speeds are 13 m s$^{-1}$ for northeasterly wind and 8 m s$^{-1}$ for southwesterly wind. We can conclude that the topographic effects cause an opposite change in the simulated wakes compared to the SCADA data. Then, we examined the wind field by WAsP CFD and the wake models in PyWake. For northeasterly wind, the wake effects weaken at the downstream wind turbine due to the acceleration of wind over the top of the hill. For southwesterly wind, the wind speed at the downstream wind turbine decreases significantly because of the weak winds in the lee of the hill and the wake extending horizontally to the downstream wind turbine from the upstream wind turbine. From the simulation results, this study demonstrated that the downstream extents of the wakes are key for obtaining results consistent with the SCADA observations. These results indicate the importance of considering the selection of wake models and topographic conditions.

By analysing the SCADA data, this study reveals the fundamental characteristics of bidirectional wake effects over complex terrain. The SCADA data have the potential to provide much information on wakes. The method used for detecting wake effects in this study demonstrates the potential to utilize the SCADA data collected during downtime of wind turbines due to maintenance and curtailment. Although this study focuses on onshore wind turbines, the findings provide important suggestions for offshore wind turbines near the coast influenced by terrain effects and for wind power plants subject to multiple wake interactions.

We further need to investigate the following aspects. First, examining the impacts of atmospheric stability on wind turbine wakes is a challenge. In the study area, northeasterly wind dominates in summer, and southwesterly wind dominates in winter (Sasanuma and Honda, 2020). That is, dominant winds and stability conditions might typically be fixed to the seasons. The following studies have investigated stability effects on wakes using numerical simulations (Uchida and Takakuwa, 2019; Yamaguchi et al., 2024) and observations (Oqaily, 2025). Uchida and Takakuwa (2019) show that the wind turbine wakes on the top of the hill follow the terrain under stable atmospheric conditions, whereas the wake rises on the lee side under neutral atmospheric stability conditions. Therefore, the extent of the wake effect depends not only on the terrain but also on the atmospheric stability. Then, increased turbulence intensity owing to wakes increases the risks of fatigue loads and shortens the lifespan of wind turbines. We have a next plan to quantitatively evaluate the fatigue loads and fatigue lifetime at the downstream wind turbine by using aeroelastic simulation. Insights into the risks of the downstream wind turbine will be valuable for the operation and control of wind power plants.

## Appendix A: List of the 12 wake models

We present 12 wake models used in this study in Table A1.

**Table A1.** Abbreviations and full names for the 12 wake models in PyWake.

| Abbreviation | Full name | Reference |
|---|---|---|
| NOJ | NOJDeficit | Jensen, 1983 |
| TurboNOJ | TurboNOJDeficit | Nygaard et al., 2020 |
| Fuga | FugaDeficit | Ott et al., 2011 |
| Bastankhah | BastankhahGaussianDeficit | Bastankhah and Porté-Agel, 2014 |
| IEA37 | IEA37SimpleBastankhahGaussianDeficit | Bastankhah and Porté-Agel, 2016 |
| Niayifar | NiayifarGaussianDeficit | Niayifar and Porté-Agel, 2016 |
| Zong | ZongGaussianDeficit | Zong and Porté-Agel, 2020 |
| Carbajofuertes | CarbajofuertesGaussianDeficit | Carbajo et al., 2018 |
| TurboGaussian | TurboGaussianDeficit | Nygaard et al., 2020 |
| GCL | GCLDeficit | Larsen, 2009 |
| Blondel2020 | BlondelSuperGaussianDeficit2020 | Blondel and Cathelain, 2020 |
| Blondel2023 | BlondelSuperGaussianDeficit2023 | Blondel, 2023 |

## Appendix B: Wind speed reduction simulated by the 12 wake models

We show the results of wind speed reduction using the 12 wake models for northeasterly wind and southwesterly wind (Figs. B1 and B2). The horizontal axis indicates differences in wind direction, with the minimum wind speed ratio at 0º for both the SCADA data and the results of the wake models. For northeasterly wind, the wake models that represent the wake well overestimate the reduction in wind speed ratio at low wind speeds (Figs. B1d, B1e, B1h, B1i, B1k, and B1l). For southwesterly wind, most of the wake models underestimate the reduction in wind speed ratio. Only the Bastankhah wake model (Fig. B2d) and the TurboGaussian wake model (Fig. B2i) show the closest results of the SCADA data below 10 m s$^{-1}$.

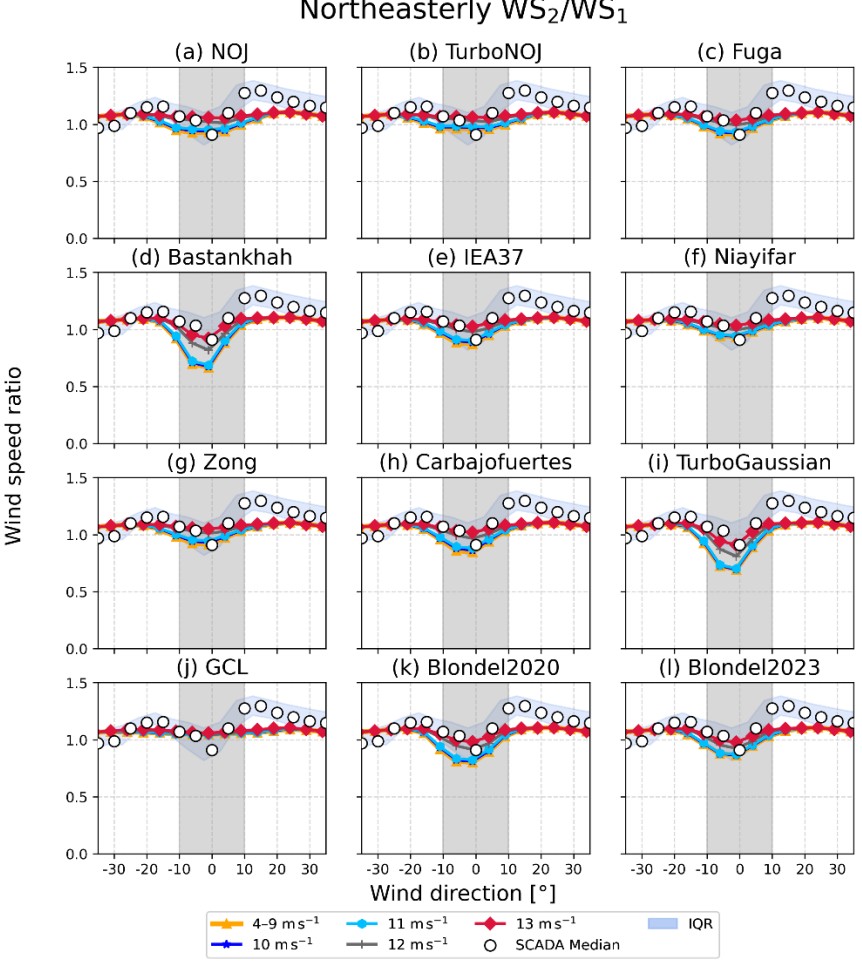

**Figure B1.** Wind speed ratio derived from the 12 wake models for northeasterly wind. Colored lines are results derived from each wake model for every 1 m s$^{-1}$, ranging from 4 to 13 m s$^{-1}$. The results ranging from 4 to 9 m s$^{-1}$ are shown by one line. The horizontal axis indicates differences in wind direction, with the minimum wind speed ratio at 0º for both the SCADA data and the results of the wake models. The blue band denotes the interquartile range (IQR) of the SCADA data, and white circles denote median values in every 5º bin. Gray shading indicates the ranges of wind direction with wake effects.

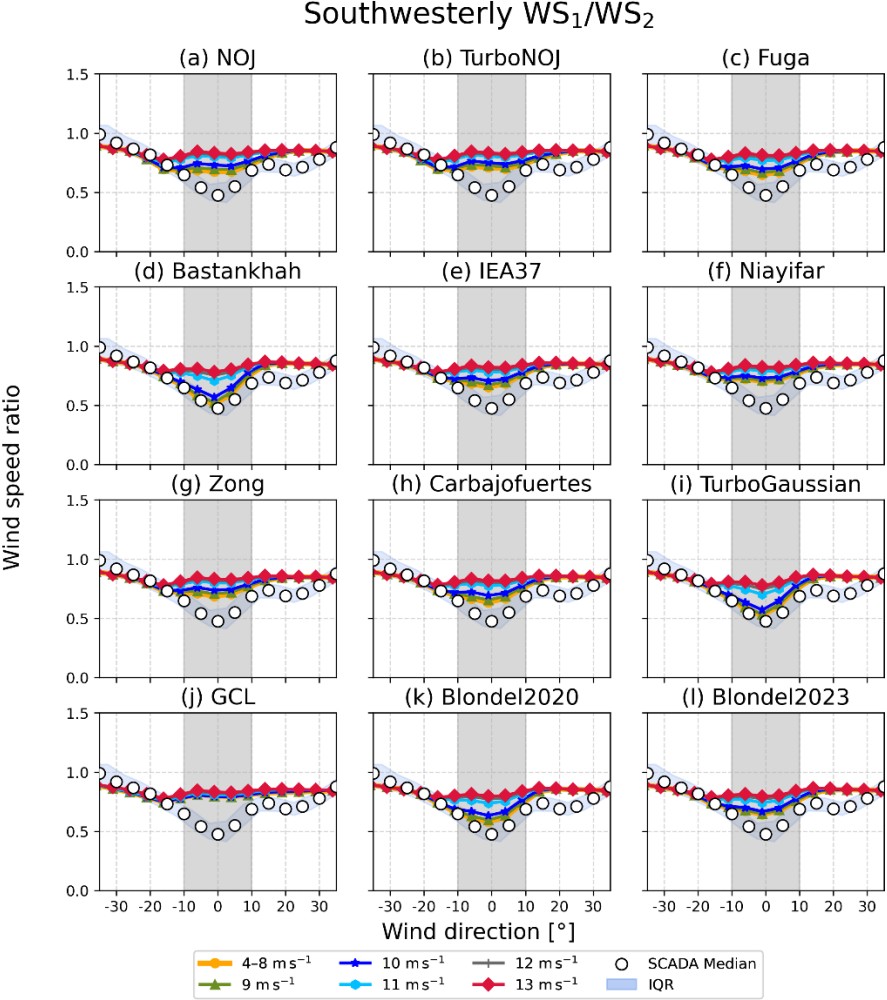

**Figure B2.** The same as in Fig. B1, but for southwesterly wind. The results ranging from 4 to 8 m s$^{-1}$ are shown by one line.

## Appendix C: Vertical structures of wakes simulated by the 12 wake models

We compare the vertical structures of wakes over complex terrain among the 12 wake models for northeasterly wind and southwesterly wind (Figs. C1 and C2). In Figs. C1a, C1b, C2a, and C2b, the wakes show linear downstream extensions and no vertical variation. The Fuga wake model exhibits the reduction in wind speed outside the rotor areas (Figs. C1c and C2c). The other

wake models produce Gaussian-shaped wakes, whereas the simulated results differ in their wake extents and the reduction in wind speed immediately behind the wind turbines. The Bastankhah wake model and the TurboGaussian wake model show that the wakes generated by the upstream wind turbines reach the rotor of the downstream wind turbines without significant attenuation (Figs. C1d, C1i, C2d, and C2i).

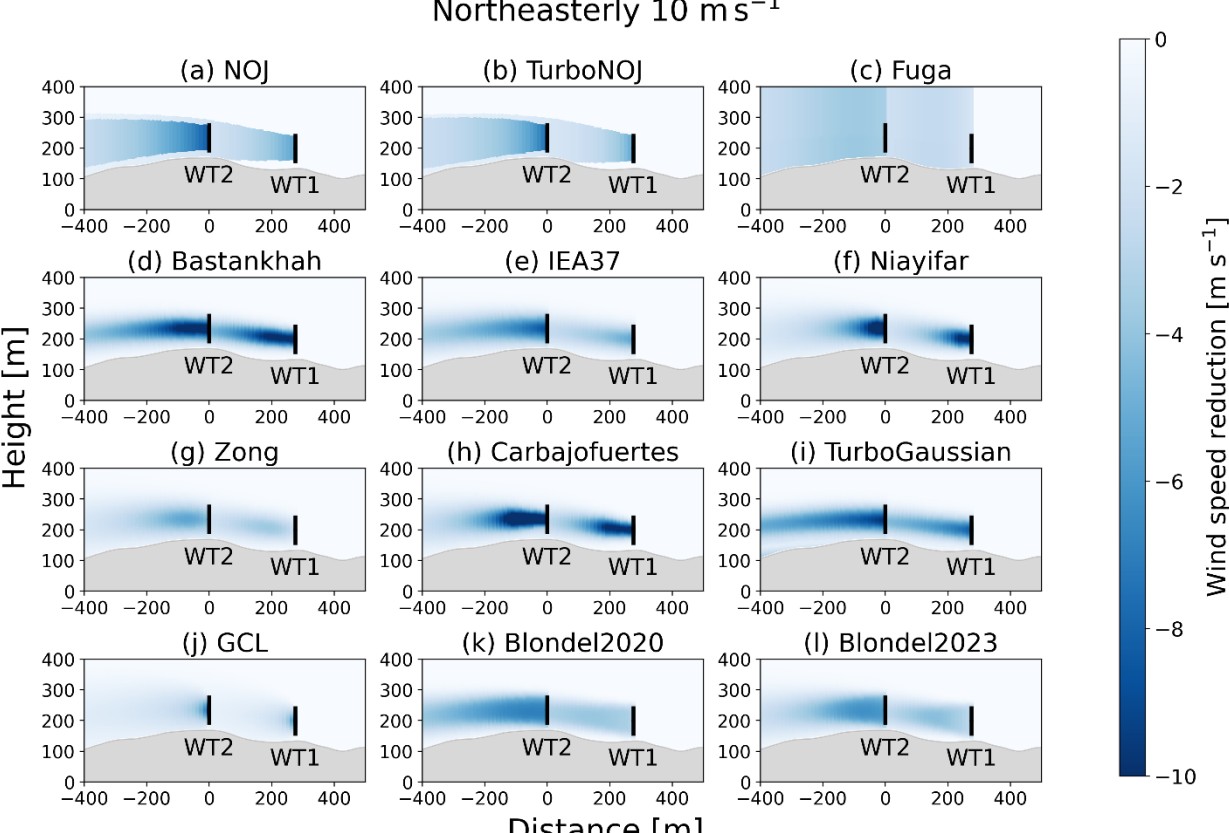

**Figure C1.** Reduction in horizontal wind speed due to the wakes simulated by the 12 wake models for northeasterly. The inflow wind speed far upstream is set to 10 m s⁻¹. The color shading represents the difference in horizontal wind speed between wake conditions and no-wake conditions. The black lines indicate the positions of the wind turbine rotors.

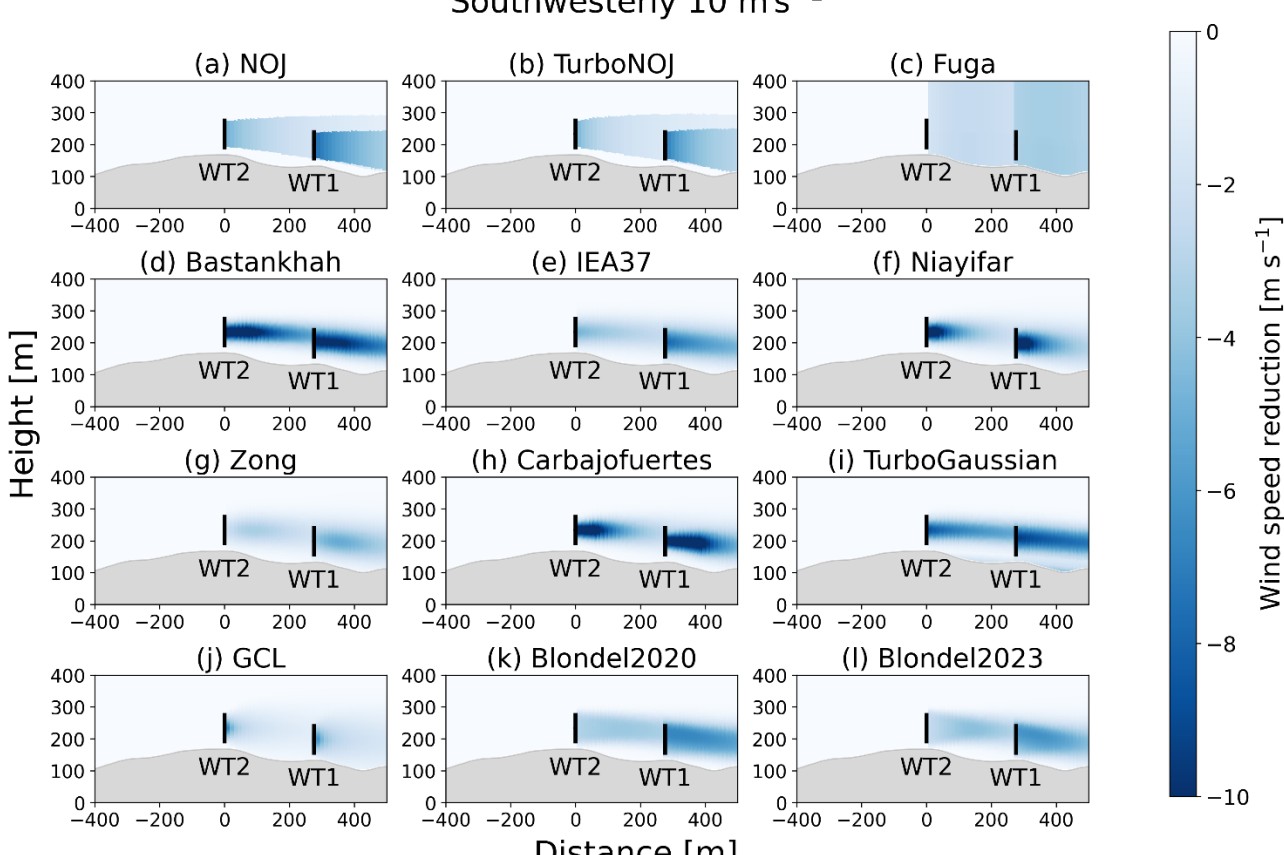

**Figure C2.** The same as in Fig. C1, but for southwesterly wind.

**Code availability**

We used an open source in PyWake 2.5 (https://topfarm.pages.windenergy.dtu.dk/PyWake/ (last access: 17 July 2025)) and WAsP 12 (https://www.wasp.dk/software/wasp-cfd (last access: 17 July 2025)) in this study.

**Data availability**

The SCADA data were provided by Tsugaru Peninsula Eco Energy Co., Ltd. and were unavailable to the public.

**Author contribution**

Writing (original draft preparation) and visualization: NS and TS. Writing (review and editing): NS, TS, AH, CB, NT, MG, and
MN. Methodology and investigation: NS, AH, MG, and TS. Software: NS and MN. Conceptualization: NS, AH, and TS. All authors contributed with critical feedback on this research and have read and agreed to the published version of the paper.

**Competing interests**

The authors declare that they have no competing interests.

## Acknowledgments

We would like to thank the anonymous reviewers and the editor for their constructive comments on this paper. We wish to express our gratitude to the staff of Tsugaru Peninsula Eco Energy Co., Ltd. and the staff of Japan Steel Works, Ltd. for providing the observational data. We thank Mr. Mads Mølgaard Pedersen for analysing PyWake and Ms. Ginka Georgieva Yankova for analysing data uncertainty. This research was supported by the JST Challenging Research Program for Next-Generation Researchers JPMSP2152.

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
