# Peer review of "Bidirectional wakes over complex terrain using SCADA data and wake models"

_Wind Energy Science, 2025_

## Author Comment (AC1)

Review of "Bidirectional wakes over complex terrain using SCADA data and wake models" by N. Sasanuma and co-authors.

This paper describes observations made using SCADA data over roughly two years of operation of two wind turbines sited on complex terrain in northern Japan. The orientation of the two turbines is roughly southwest-northeast. WT2 is slightly elevated compared to WT1. WT2 is upstream for southwesterly wind ($225^0$) and WT1 is upstream when the wind is northeasterly ($45^0$). Analysis of the wind speed ratio (wind speed at downstream turbine divided by wind speed at upstream turbine) and turbulence intensity shows that wake effects of the upstream turbine are felt on the downstream turbine only for southwesterly wind (i.e. when WT2 is upstream). Wake effects are not felt on the downstream turbine for the northeasterly wind (when WT1 is upstream). This is attributed to the effect of topography and because WT2 is at a higher elevation as compared to WT1. Twelve analytical or semi-analytical wake models available in PyWake are evaluated for their ability to predict the wake effects and show mixed results with errors ranging from under 1 % to 66 %.

Overall, the paper is well-written and easy to read. Real-field data are always scarce, and hence valuable, and this paper fills this gap. The presentation and interpretation of the results could be improved a bit. The effect of stability, which can have a non-trivial impact, is completely ignored in the paper. The section on evaluation of analytical models is quite superficial and should be improved substantially. Specific comments that the authors should address are given below.

**Author response:** We appreciate your time and effort in reviewing our manuscript and providing supportive and valuable comments. We have incorporated your suggestions into the revised manuscript. The revised parts are highlighted in the manuscript. The authors' responses to the reviewer's comments are described below. The symbol "**Author response**" means the author's responses.

Major Points:

1. The major takeaway from this study seems to be that if an upstream turbine is at a higher elevation than the downstream turbine, its wake affects the downstream turbine. If the upstream turbine is at a lower elevation than the downstream turbine, its wake does not affect the downstream turbine.

   If this above understanding is correct, the authors should try to find evidence as to whether this is supported by other observations/experiments/simulations. One possible explanation is likely in recent wind-tunnel experiments on complex terrain (Chen et al., Applied Energy, 2025. 10.1016/j.apenergy.2024.125044). This study shows that the wake of a turbine sited on a hilltop follows the terrain and bends down along with the surface. However, the wake of a turbine sited upstream of the hill does not bend upwards and follow the terrain in a similar manner.

   The authors should consider whether this aligns with their observations and include a discussion on this. Other similar observations/experiments/simulation results would also strengthen the main argument of this paper.

**Author response:** Thank you for your comments. Following your suggestion, we first added the figures of wind speed reduction for northeasterly wind (Fig. 14b) and southwesterly wind (Fig. 14d). Then, we added the description on the results of Fig. 14 and the discussion as below in the 3rd paragraph of Section 3.2. In addition, we cited the following papers to strengthen the main results of this study (Berg et al., 2017; Dar et al., 2019; Wenz et al., 2022).

  "To investigate the different wake effects between northeasterly and southwesterly winds, we show vertical cross sections of wind computed by WAsP CFD and PyWake along the connection the wind turbines (Figure 14). We computed wind flows with and without the effects of the wind turbines and derived the reduction in horizontal wind speed. In Figs. 14b and 14d, the TurboGaussian wake model is used. We can consider the path of the wake generated by the upstream wind turbine because the streamline is assumed to pass through the center of the wind turbine wake (Sesarego et al., 2020). For northeasterly wind, the wind reaches the hub height of WT2 from the hub height of WT1 due to the wind ascending the hill (Fig. 14a). Moreover, the flow accelerates on the top of the hill around the location of WT2, and the accelerated wind blows through the rotor surface of WT2. When the wind turbines exist, the Gaussian-shaped wake generated by WT1 rises slightly upward along the terrain and reaches the center of the rotor of the downstream wind turbine (Fig. 14b). However, the wake weakens at WT2 because the flow accelerates on top of the hill. Thus, the wind speed ratios are generally higher than 1.0 in no-wake conditions (Fig. 7a). For southwesterly wind, the weak winds in

the lee of the hill of WT2 cover the location of WT1 below the hub height and extend through the height of the lower part of the WT1 rotor (Fig. 14c). In contrast, the strong winds on the top of the hill reach the hub height of WT1. The resulting strong vertical wind shear at the height of the rotor of WT1 contributes to the increase in turbulence. No acceleration occurs in the lee of the hill between WT1 and WT2, and the flow separation is created by the hill. When the wind turbines exist, the wake generated by WT2 extends horizontally through the upper part of the rotor of WT1 (Fig. 14d). The wake doesn't follow the terrain line in the lee of the hill (Berg et al., 2017; Dar et al., 2019; Wenz et al., 2022). Thus, the wind speed ratios are generally lower than 1.0 in no-wake conditions (Fig. 7c)."

[Figure]

**Figure 14.** Vertical cross sections along the blue dashed line in Fig. 1 for (a) (b) northeasterly wind and (c) (d) southwesterly wind. In (a) and (c), we show horizontal wind speed (color shading) and wind vectors on the cross section with no wake effects of the wind turbines, although we show the positions of the rotors of the wind turbines by gray dashed line. In (b) and (d), we show a reduction in horizontal wind speed (color shading) due to the wind turbine wake. The data from WAsP CFD are available from 5 m.

Berg, J., Troldborg, N., Sørensen, N.N., Patton E. G., and Sullivan, P. P.: Large-Eddy Simulation of turbine wake in complex terrain, J. Phys.: Conf. Se. 854, 012003, 10.1088/1742-6596/854/1/012003, 2017.

Dar, A. S., Berg, J., Troldborg, N., and Patton, E. G.: On the self-similarity of wind turbine wakes in a complex terrain using large eddy simulation, Wind Energ. Sci., 4, 633–644, https://doi.org/10.5194/wes-4-633-2019, 2019.

Wenz, F., Langner, J., Lutz, T., and Krämer, E.: Impact of the wind field at the complex-terrain site Perdigão on the surface pressure fluctuations of a wind turbine, Wind Energ. Sci., 7, 1321–1340, https://doi.org/10.5194/wes-7-1321-2022, 2022.

2. Are atmospheric stability effects important at this site? Hypothetically, it is possible that the southwesterly wind is always accompanied by stable atmospheric conditions, under which the wake is known to persist longer, and northeasterly wind is always accompanied by unstable conditions, under which the wake recovers faster. Thus, the significant wake effects observed (not observed) for southwesterly (northeasterly) wind could hypothetically be simply due to thermal stratification. Can this explanation be ruled out using some data, or is this a valid explanation for the observations?

**Author response:** Thank you for your comments. We understand the importance of considering the atmospheric stability for wind turbine wakes. For example, the following studies have investigated stability effects on wakes using numerical simulations (Uchida and Takakuwa, 2019; Yamaguchi et al., 2024) and observations (Oqaily, 2025). Thus, we revised the discussion on stability in the last paragraph of Section 4 as below. We recognize that the effects of atmospheric stability on wakes are our next challenges.

 "In the study area, northeasterly wind dominates in summer and southwesterly wind dominates in winter (Sasanuma and Honda, 2020). That is, dominant winds and stability conditions might typically be fixed to the seasons. The studies have investigated stability effects on wakes using numerical simulations (Uchida and Takakuwa, 2019; Yamaguchi et al., 2024) and observations (Oqaily, 2025). Uchida and Takakuwa (2019) show that the turbine wake on the top of the hill follows the terrain under stable atmospheric conditions; however, the wake rises on the lee side under neutral atmospheric stability conditions. Therefore, the magnitude of the wake effect depends not only on the terrain but also on the atmospheric stability. Further research on the impact of atmospheric stability on turbine wake is necessary."

Oqaily, D. A., Giani, P., & Crippa, P.: Evaluating WRF Multiscale Wind Simulations in Complex Terrain: Insights from the Perdigão Field Campaign. Journal of Geophysical Research: Atmospheres, 130(15), e2025JD044055. https://doi.org/10.1029/2025JD044055, 2025.

Sasanuma, N., Honda, A.: Wind analyzing of wind turbines and lighthouse in Tappi with relative point of view, Wind Engineering Symposium, 26, 19-24, https://doi.org/10.14887/natsympwindengproc.26.0_19, 2020.

Uchida, T., and Takakuwa, S.: Numerical Investigation of Stable Stratification Effects on Wind Resource Assessment in Complex Terrain. Energies, *13*(24), 6638. https://doi.org/10.3390/en13246638, 2019.

Yamaguchi, A. Tavana, A. and Ishihara, T.: Assessment of Wind over Complex Terrain Considering the Effects of Topography, Atmospheric Stability and Turbine Wakes. Atmosphere, 15, 723. https://doi.org/10.3390/atmos15060723, 2024.

3. The section on wake model evaluation lacks many details. For example,

(a) what is the inflow provided to the models? Is it a uniform in the vertical, or a sheared profile?

**Author response:** We set the inflow wind speed for far upwind. In this case, the wind speed is 10 m s$^{-1}$ at 100m height, and roughness length $z_0$ is 0.0032m. Therefore, the inflow wind speed has a slightly sheared profile. We added these descriptions to Section 2.6.

(b) Line 91 mentions 'bush trees' on the terrain. Is the effect of these trees incorporated into the wake models through, e.g., a canopy displacement height, or through an aerodynamic roughness length?

**Author response:** WAsP CFD sets the roughness length depending on landscape type. The roughness length in the study area ranges from 0.1 to 1.0 m. We added these descriptions to Section 2.6.

(c) Each of the 12 wake models have at least one (probably more) tunable parameters which can drastically change their predictions. The authors have not mentioned what values were assigned to these parameters. If some 'default' values were used, those should be mentioned. Also, it is crucial to mention sensitivity of the predictions to these parameters.

**Author response:** Thank you for your suggestions.
- We use the "default" values of the parameters, and tuning the parameters is not scope of this study. We mentioned this in Section 1.
- As you indicated, the sensitivity of the parameters of the wake model is important. We agree with this point. However, the sensitivity analysis is beyond the scope of this study. Sensitivity analysis of the parameters of the 12 wake models is a subject for further study.

(d) Is there any particular reason that the 'Bastankhah' and 'TurboGaussian' models are more accurate than the others?

**Author response:** For 'Bastankhah' and 'TurboGaussian' wake models, thrust coefficient (Ct) and wake width ($\sigma$) mainly affect the reduction in wind speed. The maximum reduction in wind speed for both wake models can be expressed by the following equation.

$$C(x) = 1 - \sqrt{1 - C_T/8(\sigma(x)/d_0)^2} \ (1)$$

where $C(x)$ is a maximum decrease in wind speed at the downwind distance (x), $C_t$ is the thrust

coefficient, $\sigma(x)$ is wake width at the downwind distance (x), and $d_0$ is the diameter of the wind turbine. Other wake models, except Fuga and GCL wake models, use equation (1) generally. For wake width, Bastankhah wake model uses equation (2) and TurboGaussian wake model uses equation (3) respectively. 'Bastankhah' and 'TurboGaussian' wake models commonly use ε using equation (4) and Ct effects for both reduction in wind speed and wake width.

$$\frac{\sigma}{d_0} = kx/d_0 + \varepsilon \ (2)$$

$$\frac{\sigma}{d_0} = \varepsilon + \frac{AI_0}{\beta}\left(\sqrt{\left(\alpha + \frac{\beta x}{D}\right)^2 + 1} - \sqrt{1 + \alpha^2}\right) - \ln\left(\left[\frac{\left(\sqrt{\left(\alpha + \frac{\beta x}{D}\right)^2 + 1} + 1\right)\alpha}{\sqrt{1 + \alpha^2}\left(\alpha + \frac{\beta x}{D}\right)}\right]\right) \ (3)$$

$$\varepsilon = 0.2\,\beta \ (4)$$

$$\beta = 1/2\left(\frac{1 + \sqrt{1 - C_T}}{\sqrt{1 - C_T}}\right) \ (5)$$

where D is the rotor diameter of the downstream wind turbine, $\alpha = c_1 I_0$ (c1=1.5), $I_0$ is the free stream turbulence intensity.

Thus, we speculate that "Bastankhah" and "TurboGaussian" wake models have a large sensitivity to the thrust coefficient compared with the other wake models. We understand that it is our next challenge to investigate the reason why these two models show the results consistent with the observations.

Bastankhah, M., and Porté-Agel, F.: A new analytical model for wind-turbine wakes. Renewable Energy, 70, 116-123. https://doi.org/10.1016/j.renene.2014.01.002, 2014.

Pedersen, M. M., van der Laan, P., Friis-Møller, M., Forsting, A. M., Riva, R., Romàn, L. A. A., Risco, J., C., Quick, J., Christiansen, J. P. S., Olsen B. T., Rodrigues, R. V., and Réthoré, P. E., DTU WindEnergy/PyWake: PyWake (v2.5.0), https://doi.org/10.5281/zenodo.6806136, 2023.

(e) The two models seem to be performing better when the wind speed is set to a certain value. However, wake models are generally agnostic to wind speed since they predict normalized velocity deficits. The only way wind speed enters into a wake model is through the thrust coefficient that gets modulated with wind speed. Is this the reason for the two models to be performing better in certain wind regimes than others?

In view of the above points, perhaps it would be better to focus on a smaller number of models more thoroughly than to superficially show results of a dozen models.

**Author response:** As we described the answer in (d)**,** "Bastankhah" and "TurboGaussian" wake models have a large sensitivity to the thrust coefficient. We recognize that this point merits further study, as you indicated. However, the purpose of this study is to compare the reproducibility of wake models over complex terrain between wake models and to show the terrain effects systematically modify the simulated wakes.

Minor Points:

1. Is Fig 1a really needed? It should be sufficient to mention latitude and longitude of the site of study. Is there anything specific that is being conveyed by map at this scale?

   **Author response:** Thank you for your suggestion. Following your suggestion, we have deleted Figure 1a, as Figure 1b is sufficient to show the locations of wind turbines and the surrounding terrain.

2. Similarly, I am not sure if the photograph in Fig. 2a is really necessary.

   **Author response:** We believe that Figure 2a is important for easily understanding the problem setting of this study and the surrounding environment of the wind turbines and is supportive of Figure 2b. Therefore, we keep Figure 2a.

3. The difference in the elevations between the two turbines seems to be 0.44D ((169-132)/83.3) rather than 0.5D. This should be mentioned precisely rather than rounding off to 0.5D.

   **Author response:** We corrected the difference in elevation between the two turbines to 0.44D from 0.5D.

4. Appendix A used to correct wind direction and wind speed due to turbine rotation is not clear. Lines 385-395 should be expanded and the method should be explained in more detail. Are there any previous references that can be cited for this?

   **Author response:** After consideration, we decided not to correct the deviations of wind direction and wind speed due to the wind turbine rotation. Therefore, we deleted Appendix A. (In this study, we discuss the maximum wake effects from the minimum wind speed ratio in Figure 6, 7, 11, and 12 without correcting the systematic offset in wind direction (Mittelmeier et al. (2017)).

   Mittelmeier, N., Blodau, T., and Kühn, M.: Monitoring offshore wind farm power performance with SCADA data and an advanced wake model, Wind Energ. Sci., 2, 175–187, https://doi.org/10.5194/wes-2-175-2017, 2017.

5.  Since the 'Bastankhah' and 'TurboGaussian' models seem to be performing the best, they should be explained at least briefly, maybe in an appendix. Also, please provide a reference on line 164.

**Author response:** Thank you for your comments. We added an explanation and references of the two wake models to Section 2.6.

6.  The paragraph around line 150 is inconsistent. North-easterly wind implies WT1 is upstream and WT2 is downstream, and wind speed ratio is defined as WS2/WS1. This means the wind speed ratio is defined as the ratio of wind at downstream turbine to that at upstream turbine. The first sentence of this paragraph states the opposite. Please clarify this.

**Author response:** Thank you for pointing that out. Your suggestion is correct, and we corrected the first sentence below.
"To analyse mutual wake effects between the two wind turbines, we define a wind speed ratio of wind speed at the downstream wind turbine to that at the upstream wind turbine." Additionally, we have included Table 1 to clearly define the wind speed ratio based on the combination of wind direction and wake condition.

7.  Line 155: It is slightly awkward to say that computational models are used to validate the observations. It should be the other way around. Perhaps 'consistency check' or something similar would be a better phrase here.

**Author response:** We have corrected the first sentence of section 2.6 as below. "We utilized Wind Atlas Analysis and Application Program Computational Fluid Dynamics (WAsP CFD; Bechmann, 2015) in combination with PyWake (Pedersen et al., 2018) to closely examine the the observed wind and wakes over complex terrain."

8.  Line 165: Is the power curve of the actual turbine model not available, and can't it be implemented into the PyWake code? How do the power curves of the J82-2.0 and Vestas V80 turbines compare with each other?

**Author response:** Due to confidentiality concerns, we are not able to show the absolute power curve here. However, we confirmed that the power curves from V80 and the power output from the SCADA data are consistent with each other below the rated wind speed (Figure R1).

[Figure]

[Figure]

Figure R1. Power curve of V80 wind turbine (blue curve). Black, gray, and white dots are the power output from the SCADA data of the wind turbine in the study site. The power output is divided by the rated power output.

9. 5 can be augmented with $C_P$ of the V80 turbine for completeness.

   **Author response:** Unfortunately, PyWake does not disclose the power coefficient of V80. PyWake computes power output directly from the power curve table.

10. 7(a) and 7(c) deal with no-wake conditions. Here, there is an average ratio of 1.2 in 7(a) and of a little below 1 in 7(c). This speedup/slowdown is entirely because of terrain effects. Can this be explained using, say, the WAsP simulation described later?

    **Author response:** Thank you for pointing this out. We added the sentence to show the consistent wind speed ratio under no-wake conditions between SCADA data and WAsP CFD to Section 3.2 as below.

    "For northeasterly wind, the wind reaches the hub height of WT2 from the hub height of WT1 due to the wind ascending the hill (Fig. 14a). Moreover, the flow accelerates on the top of the hill or around the location of WT2 and the accelerated wind blows through the rotor surface of WT2. …. Thus, the wind speed ratios are generally higher than 1.0 in no-wake conditions (Fig. 7a). "
    "For southwesterly wind, the weak winds in the lee of the hill of WT2 cover the location of WT1 below the hub height and extend through the height of the lower part of WT1 rotor (Fig. 14c). In contrast, the strong winds on the top of the hill reach the hub height of WT1. The resulting strong vertical wind shear at the height of the rotor of WT1 contributes to the increase in turbulence. No acceleration occurs in the lee of the hill between WT1 and WT2,

and the flow separation is created by the hill. ….Thus, the wind speed ratios are generally lower than 1.0 (Fig. 7c). ”

11. Labels (a), (b) etc can be reduced in size in almost all figures.

**Author response:** We adjusted the size and thickness of the labels (a), (b) etc.

12. 7: How are 'no-wake' and 'wake' conditions identified? Why is there no wake at certain time instants? Is it because the upstream turbine is not operating, and measurements at such time instants are called 'no-wake'? Or is the 'no-wake' condition due to other effects such as lateral deflection of the wake due to yaw misalignment? Where does the wake 'go' under 'no-wake' conditions?

**Author response:** We identify "no-wake conditions" and "wake conditions" only by the operating state of the upstream wind turbine. The wake in this study means a wake induced by the rotor rotation of the upstream wind turbine. Therefore, "No-wake conditions" mean that the upstream wind turbine is not operating. We mentioned this method in Section 2.4.

13. Line 213: Minor typo: "To exam the maximum wake effects…"

**Author response:** Corrected "To examine the maximum wake effects, ……."

14. Are the lines corresponding to turbulence levels A and A+ (Fig. 9) described by an analytical expression? If so, can these expressions be provided?

**Author response:** We added the following equations to 4th paragraph of Section 3.1 to show the turbulence intensity (TI) defined by International Electrotechnical Commission: IEC 61400-1 (IEC, 2019). WS is wind speed.

TI_category A+ = 0.18 (0.75 + 5.6 / WS) (1)
TI_category A = 0.16 (0.75 + 5.6 / WS) (2)

IEC – International Electrotechnical Commission: IEC 61400-1: Wind energy generation systems – Part 1: Design requirements, 4th Edn., Geneva, Switzerland, 2019.

15. 9(a) and 10(a) are both for northeasterly wind, when WT1 is upstream and WT2 is downstream. The wind speed at WT2 (i.e. x axis) in Fig. 9(a) goes till 18 m/s while in Fig. 10(a) it goes till only 6 m/s. Are these plots consistent with each other?

**Author response:** We use the different wind speeds for the x-axis in Fig. 9 and the x-axis in Fig. 10. In Figure. 9, we use the wind speed of the downstream wind turbine on the x-axis to examine the wake effects due to the rotation of the upstream wind turbine. In Figure 10, we use the wind speed of the upstream wind turbine on the x-axis to examine the reduced output at the downstream wind turbine compared with the output at the upstream wind turbine. We mentioned these points in the captions of Figures 9 and 10.

16. 10: what is the power normalized with?

**Author response:** The normalized power output in Figure 10 is the power output divided by the rated power output. We added an explanation to the caption of Figure 10.

17. In Appendix B or Table 1, references for the 12 models should be provided.

**Author response:** We added the reference papers for each wake model in Table A1 in Appendix A.

18. Why is the WaSP simulation not performed along the same line joining the turbines?

**Author response:** We derived WAsP CFD simulation along the line joining the two wind turbines to remake Figure 14.

19. Line 358: "Additional effects of topography to the wake effects cause opposite changes in the simulated wakes." It is unclear what this line means. Where has this been shown in the paper?

**Author response:** We revised the description as below.
"In other words, topographic effects cause opposite changes in the simulated wakes compared to the SCADA data."

---

## Author Comment (AC2)

Wind Energy Science wes-2025-130
Responses to Reviewer 1

This paper addresses a practical topic of bidirectional wake effects in complex terrain using SCADA data and wake modeling. Comparative studies contributed to the selection of wake models in complex terrain. The following comments are intended to help strengthen the manuscript for potential publication.

**Author response:** We appreciate your time and effort in reviewing our manuscript and providing supportive and valuable comments. We have incorporated your suggestions into the revised manuscript. The authors' responses to the reviewer's comments are described below. The symbol "**Author response**" means the author's responses.

**1. Abstract:**

The abstract lists the work and results but does not clearly articulate the motivation and contributions of the study. It is recommended that the authors restructure the abstract to begin with a broader background, narrow down to the specific focus on bidirectional wake effects, and end with a stronger conclusion that clearly states the novelty and significance of the work.

**Author response:** As recommended by the reviewer, we revised the abstract as follows.
- [The 2nd sentence of the abstract]
  We added the sentence below to explain the background and motivation of this study.
  "The extent to which complex terrain affects wake behavior has not yet been fully studied."
- [The last sentence of the abstract]
  We added the sentence below to enhance the novelty and significance of our work.
  "This comparative study contributes to understanding the additional effects of topography on wake effects in onshore wind power plants and offshore wind power plants near the coast."

**2. Article Structure:**

The overall structure of the paper could be reorganized; for example, lines 54-63 seem more appropriate in the methodology section rather than the introduction.

**Author response:** Thank you for this suggestion. Following your suggestion, we moved the descriptions on the study site to Section 2.1 and the descriptions on wind climate to Section 2.3. In the fourth paragraph of Section 1, we only mention the results of Sasanuma and Honda (2022; 2024) to review the previous studies.

Additionally, some subheadings could be more informative. For instance, a title such as "2.1 Two Turbines" is too generic, and "2.2 SCADA" does not concisely describe the content of the section.

**Author response:** Thank you for your suggestion. We revised the subheadings in Sections 2.1, 2.2, 2.3, and 2.6 to be more informative and clearer as below.

- 2.1 "Study site and two wind turbines" from "Two wind turbines".
- 2.2 "SCADA data at the two wind turbines" from "SCADA".
- 2.3 "Wind climate" from "Wind farm climate".
- 2.6 "Flow simulations with wake models" from "Flow wake models".

**3. Introduction:**

The background does not effectively introduce the primary object of the study. Moreover, literature reviews were unable to identify the progress and the key research gaps of the research. It is recommended to supplement the review with more recent and relevant work.

**Author response:** Thank you for your suggestion. Following your suggestion, we fully revised the fourth paragraph of Introduction (Section 1) to cite and review the previous and recent studies that focus on wind turbine wakes over complex terrain. In response to comment 6 of Reviewer 1, we listed the studies cited in the paragraph. Owing to the revision, we could clarify the key research gaps and the scope of our paper and highlight the fact that focusing on bidirectional wakes over complex terrain is a new approach.

**4. Methods:**

The theoretical framework is unclear. The authors defined the "wind speed ratio" and conducted analyses based on it. It is recommended to provide a mathematical formula and a detailed explanation.

**Author response:** Thank you for your suggestions.

- Following your suggestion, we added the mathematical formula $WS_2 / WS_1$ for northeasterly wind and $WS_1 / WS_2$ for southwesterly wind to Section 2.5, where $WS_1$ is wind speed at WT1 and $WS_2$ is that at WT2.
- In addition, we added Table 1 to Section 2.5 to clearly show the definitions of wind speed ratio according to the combination of wind direction and wake condition.

- In Section 2.2, we also added a mathematical formula to define turbulence intensity as $\sigma$ / WS, where $\sigma$ is the standard deviation of wind speed, and WS is the 10-minute mean wind speed.

Table 1. Definition of wind speed ratio according to the combination of wind direction and the presence or absence of wake. White shading indicates the condition that the upstream wind turbine is not in operation, and gray shading indicates the condition that the upstream wind turbine is in operation.

|  | Northeasterly wind | Southwesterly wind |
|---|---|---|
| No-wake conditions | $WS_2 / WS_1$ | $WS_1 / WS_2$ |
| Wake conditions | $WS_2 / WS_1$ | $WS_1 / WS_2$ |

**5. Validation and Analysis:**

For this wind field test, using wake models to validate the observed SCADA data seems unreasonable.

**Author response:** Thank you for pointing this out. We gave the following two responses.

1) The previous studies show that applying the wake models to the wind flow over terrain is valid. We mentioned the previous studies below in Section 2.6. Fleming et al. (2020) investigate wake steering for onshore turbines using engineering flow calculation tool with Bastankhah model. Ruisi and Bossanyi (2019) indicate consistent reduction in wind speed due to the wind turbine wake between Bastankhah wake model and observations in an onshore wind farm. zum Berge et al. (2024) evaluated the performance of TurboGaussian wake model using a flight measurement for wind farm clusters in offshore sites. We found that these two wake models (TurboGaussian and Bastankhah models) represent wake effects more consistent with the observations than other wake models. Fischereit et al. (2022) indicate that three wake models (NOJ model, Bastankhah model, and Zong model) accurately simulate the intra-farm wakes. Jeon et al. (2015) verify the prediction accuracy of several wake models, including Jensen and GCL wake models, and found that Jensen wake model is the best for reduction in wind speed due to the wind turbine wake and GCL wake model is relatively accurate for the width of wake flow in an onshore wind farm.

2) Currently, few studies have examined the difference in performance between all the wake models in PyWake in complex terrain. Therefore, we focus on the comparison of the wake models to provide helpful information for selecting wake models in complex terrain. Although the results from some wake models are far from the SCADA results, our scope in this study is not to calibrate the wake models by tuning the parameters to the SCADA results, but to compare the simulated wakes by 12

wake models in default settings. We mentioned the description above in the last paragraph of Section 1.

Farrell, A., King, J., Draxl, C., Mudafort, R., Hamilton, N., Bay, C. J., Fleming, P., and Simley, E.: Design and analysis of a wake model for spatially heterogeneous flow, Wind Energ. Sci., 6, 737–758, https://doi.org/10.5194/wes-6-737-2021, 2021.

Fischereit, J., Schaldemose Hansen, K., Larsén, X. G., van der Laan, M. P., Réthoré, P.-E., and Murcia Leon, J. P.: Comparing and validating intra-farm and farm-to-farm wakes across different mesoscale and high-resolution wake models, Wind Energ. Sci., 7, 1069–1091, https://doi.org/10.5194/wes-7-1069-2022, 2022.

Fleming, P., King, J., Simley, E., Roadman, J., Scholbrock, A., Murphy, P., Lundquist, J. K., Moriarty, P., Fleming, K., van Dam, J., Bay, C., Mudafort, R., Jager, D., Skopek, J., Scott, M., Ryan, B., Guernsey, C., and Brake, D.: Continued results from a field campaign of wake steering applied at a commercial wind farm – Part 2, Wind Energ. Sci., 5, 945–958, https://doi.org/10.5194/wes-5-945-2020, 2020.

Jeon, S., Kim, B., and Huh, J.: Comparison and verification of wake models in an onshore wind farm considering single wake condition of the 2 MW wind turbine. Energy, 93, 1769-1777. https://doi.org/10.1016/j.energy.2015.09.086, 2015.

Ruisi, R. and Bossanyi, E.: Engineering models for turbine wake velocity deficit and wake deflection. A new proposed approach for onshore and offshore applications, in: Journal of Physics: Conference Series, vol. 1222, p. 012004, IOP Publishing, 2019.

zum, Berge, K., Centurelli, G., Dörenkämper, M., Bange, J., and Platis, A.: Evaluation of Engineering Models for large-scale cluster wakes with the help of in situ airborne measurements. Wind Energy, 27(10), 1040-1062. https://doi.org/10.1002/we.2942, 2024.

Similarly, the use of a CFD approach in Section 3.2 as a supplementary analysis of bidirectional wake effects is not fully convincing, as the CFD model itself has not been sufficiently validated for this application.

**Author response:** Thank you for pointing this out. WAsP CFD is a CFD model integrated into WAsP and is designed for simulating winds over complex terrain. (WAsP has limitations in simulating wind over complex terrain.) WAsP CFD has been used by the following studies, and the resulting wind fields over complex terrain have been analyzed and validated. Thus, WAsP CFD simulation is an effective way to study the flow and wake in complex terrain.

Bechmann, A., N. N. Sørensen, J. Berg, J. Mann, and Réthoré P.-E.: The Bolund Experiment, Part II: Blind Comparison of Microscale Flow Models, Boundary-Layer Meteorology 141 (2): 245–71, https://doi.org/10.1007/s10546-011-9637-x, 2011.

Bechmann, A.: Perdigão CFD Grid Study, DTU Wind Energy E 0120, 2016.

Sharma, P. K., Warudkar, V., & Ahmed, S.: Application of a new method to develop a CFD model to analyze wind characteristics for a complex terrain. Sustainable Energy Technologies and Assessments, 37, 100580. https://doi.org/10.1016/j.seta.2019.100580, 2020.

Troen, I., and Hansen, B. O.: Wind Resource Estimation in Complex Terrain: Prediction Skill of Linear and Nonlinear Micro-Scale Models, Paper presented at AWEA Windpower Conference & Exhibition, Orlando Orange County Convention Center, United States. May 18, 2015.

The manuscript would benefit from a thorough proofread to correct grammatical errors (e.g., "is critical issue" should be "is a critical issue" in the introduction). Attention should also be paid to improving sentence structure and logical coherence; in the same paragraph, shifts in voice (active vs. passive) and subject detract from readability.

**Author response:** We carefully checked the grammatical errors and sentence structures throughout the text. We paid attention to the use of active and passive voice and the shift of subject in the same paragraph.

---

## Author Response (AR2)

Dear Dr. Xiaolei Yang, Handling Associate Editor of Wind Energy Science

With these responses to the associate editor, we submit an original research article entitled "Bidirectional wakes over complex terrain using the SCADA data and wake models" by Sasanuma, N., Honda, A., Bak, C., Troldborg, N., Gaunaa, M., Nielsen, M., and Shimada, T.

We have revised the manuscript carefully according to the associate editor's comment. The authors' responses to the associate editor's comment are described below. The symbol "**Author response**" means the author's response.

We have thoroughly checked the manuscript for wording and clarity and made the necessary revisions. We have corrected the minor error in the plot of the wind roses in Figure 3 and related numerical values. However, these corrections do not affect the results or the conclusions of this study.

We thank the reviewers for carefully reading our manuscript and the associate editor for providing constructive comment and overseeing the review process of our manuscript.

Sincerely,

Nanako Sasanuma
Graduate School of Science and Technology, Hirosaki University

The referees' comments have been satisfactorily addressed by the authors. Before the manuscript can be accepted for publication, in Appendix A, it is suggested to add brief descriptions for the 12 wake models tested in the authors' work, with necessary expressions, values of model parameters, and explanations on whether or not the effects of complex terrain are accounted for in the model and how (if yes).

**Author response:** We appreciate your time and effort in reviewing our manuscript and providing supportive and valuable comment. We have incorporated your suggestions into the revised manuscript. Considering your comment, our responses are summarized as follows.

- To clarify the differences among the 12 wake models, we added a new paragraph at the end of Section 3.2 and Appendix C as below. In the paragraph, we summarize the reproducibility of the 12 wake models based on Figs. 11, 12, 13, B1 B2, C1, and C2. In Appendix C, we present and compare the vertical structures of the wakes simulated by the 12 wake models over the complex terrain for northeasterly wind and southwesterly wind. Moreover, we revised the descriptions of the wake model results in Appendix B.

- We use the default parameter values for all wake models available in PyWake 2.5. The main parameters that govern the reduction in wind speed are thrust coefficient ($C_t$), wake width ($\sigma$), wake expansion rate ($k$), and turbulence intensity (TI). However, the wake models use more parameters in their formulations, and the equations would be necessary to show the parameters. Thus. We considered that listing the values of all parameters in a single appendix would be beyond its intended scope.

- Instead of listing the parameters in the wake models, we considered it more beneficial to compare the simulation results of the 12 wake models. We added Appendix C to show the difference in vertical structures of the wakes simulated by the 12 wake models. A clear difference can be observed even when the default parameter values are used. This fact suggests that the differences in fundamental formulation or concepts of the wake models are important.

(The last paragraph of Section 3.2)

  From the results obtained so far, we consider the differences among the wake models. The degree of wake reproducibility depends on inflow wind speed (Figs. 11, B1, and B2). Terrain effects induce a systematic bias in the reproducibility of wakes (Figs. 12, 13, B1, and B2). From the perspective of wake structure, the differences among the wake models lie in the downstream extents of the wakes and the magnitude of the wind speed reduction immediately behind the wind turbine (Figs. C1 and C2). In particular, the downstream extents of the wakes are key for obtaining results consistent with the SCADA observations in this study. Although the simulation results might depend on parameter settings of the wake models, the present results suggest that the major differences arise from the model formulation. These results highlight the importance of selecting appropriate wake models and considering topographic situations.

(Appendix C)

**Appendix C: Vertical structures of wakes simulated by the 12 wake models**

We compare the vertical structures of wakes over complex terrain among the 12 wake models for northeasterly wind and southwesterly wind, respectively (Figs. C1 and C2). In Figs. C1a, C1b, C2a, and C2b, the wakes show linear downstream extensions and no vertical variation. The Fuga wake model exhibits the reduction in wind speed outside the rotor areas (Figs. C1c and C2c). The other wake models produce Gaussian-shaped wakes, whereas the simulated results differ in their wake extents and the reduction in wind speed immediately behind the wind turbines. The Bastankhah wake model and the TurboGaussian wake model show that the wakes generated by the upstream wind turbines reach the rotor of the downstream wind turbines without significant attenuation (Figs. C1d, C1i, C2d, and C2i) and that the simulated results reasonably capture the observed results.

[Figure]

**Figure C1.** Reduction in horizontal wind speed due to the wakes simulated by the 12 wake models for northeasterly. The inflow wind speed far upstream is set to 10 m s⁻¹. The color shading represents the difference in horizontal wind speed between wake conditions and no-wake conditions. The black lines indicate the positions of the rotors of the wind turbines. The data from WAsP CFD are available from 5 m above ground level.

Southwesterly 10 m s$^{-1}$

[Figure]

**Figure C2.** The same as in Fig. C1, but for southwesterly wind.

---

## Author Response (AR3)

Wind Energy Science wes-2025-130
Responses to Reviewers

Dear Dr. Sandrine Aubrun, Handling Chief Editor, and Dr. Xiaolei Yang, Handling Associate Editor of Wind Energy Science

With these responses to the associate editor, we submit an original research article entitled "Bidirectional wakes over complex terrain using the SCADA data and wake models" by Sasanuma, N., Honda, A., Bak, C., Troldborg, N., Gaunaa, N., M., Nielsen, M., and Shimada, T.

We have thoroughly checked the manuscript for wording and clarity and made the necessary revisions. However, these corrections do not affect the results or the conclusions of this study.

We thank the reviewers for carefully reading our manuscript, and the chief editor and the associate editor for overseeing the review process of our manuscript.

Sincerely,

Nanako Sasanuma
Graduate School of Science and Technology, Hirosaki University